# Large impact of tiny model domain shifts for the Pentecost 2014 MCS over Germany

**Christian Barthlott and Andrew I. Barrett**

Institute of Meteorology and Climate Research (IMK-TRO), Karlsruhe Institute of Technology (KIT), Karlsruhe, Germany

**Correspondence:** Christian Barthlott (christian.barthlott@kit.edu)

**Abstract.** The mesoscale convective system (MCS) that affected Germany at Pentecost 2014 (9 June 2014) was one of the most severe for decades. However, the predictability of this system was very low as the operational deterministic and ensemble prediction systems completely failed to predict the event with more than 12 hours lead time. We present hindcasts of the event using the COnsortium for Small-scale MOdeling (COSMO) model at convection-permitting (2.8 km) resolution on a large ($1668 \times 1807$ km) domain. Using this large domain allowed us to successfully simulate the whole life cycle of the system originating from the French Atlantic coast. However, even with the large domain the predictability of the MCS is low. Tiny changes to the model domain produced large changes to the MCS, removing it completely from some simulations. To demonstrate this we systematically shifted the model domain by just one grid point in eight different directions, from which three did not simulate any convection over Germany. Our analysis shows that there were no important differences in domain-averaged initial conditions nor in the preconvective environment ahead of the convective system. The main reason that one-third of these seemingly identical initial conditions fails to produce any convection over Germany seems to be the proximity of the track of the initial convective system to the coast and colder sea surface. The COSMO model simulates small horizontal displacements of the precursors of the MCS which then determine if the cells dissipate close to the sea or reach a favorable area for convective development over land and further evolve into an MCS. This study demonstrates the potentially huge impact of tiny model domain shifts on forecasting convective processes in this case, which suggests that the sensitivity to similarly small initial condition perturbations could be a helpful indicator of days with low predictability and should be evaluated across other cases, model and weather regimes.

## 1 Introduction

An accurate forecast of deep moist convection is of great societal and economic relevance due to multiple risks from heavy precipitation, strong winds, lightning, or hail. Convection-permitting models have provided a step-change in rainfall forecasting and are used operationally in many parts of the world (Clark et al., 2016). Although progress has been made through higher grid spacing of numerical weather prediction models and better parameterizations of physical processes, quantitative forecasting of convective storms remains a challenge. All forecast centers still suffer from so-called forecast busts (Rodwell et al., 2013), in which a large drop in performance occurs and the forecast skill becomes very low.

The 2014 Pentecost storms over Germany were also partly characterized by a low forecast skill. Following a period of hot weather, a series of convective systems occurred over northwestern Germany leading to significant damages with even six fatalities. The major event took place on Pentecost Monday (9 June 2014) where a mesoscale convective system originating over France traveled across Belgium and hit northwestern Germany in the evening (Mathias et al., 2017). At the German Weather Service, both the deterministic run and all 20 members of the ensemble prediction system failed to predict any severe storms over northern Germany. These events and their poor prediction motivated the study of Barthlott et al. (2017), in which several methods of improving COnsortium for Small-scale MOdeling (COSMO) model simulations were evaluated, including: a larger model do-

main, higher grid spacing, a more sophisticated microphysics scheme and different initialization times. A series of different numerical simulations for the convective events of 9 June 2014 and the previous day were performed with the COSMO model, the main findings were

- The COSMO model (in quasi operational set-up, without data assimilation) initialized at 00:00 UTC did not reproduce the mesoscale convective system (MCS) on 9 June.

- Enlarging the model domain towards the West improved the precipitation forecast only over France, due to better resolving the initiation and development of deep convection over western France and, later, secondary initiation over northern France. The MCS over Germany, however, was not simulated even with this larger domain.

- Improving both vertical and horizontal grid spacing (highest resolution 1 km) had only minor effects on the simulation results.

- An increased or reduced initial soil moisture had significant effects on the energy balance of the surface (see e.g., Barthlott et al., 2011), but still no MCS-like system was simulated over Germany.

- Although weaker than observed, later initialization times (03:00 UTC, 06:00 UTC) produced deep convection over Germany due to outflow triggering and secondary cell initiation.

Specific reasons for the model failure of the 00:00 UTC run remained unclear, but the analysis of convection-related variables indicated too high values of convective inhibition (CIN) in northern Germany. As was pointed out by Groenemeijer (2014), extending the model domain to the west and south would potentially allow storms to be captured earlier by the model. By enlarging the domain 300 km to the west, the direction from which most severe thunderstorms arrive, the lead time could be increased by 3 h (assuming a system moving with $100 \, \mathrm{km \, h^{-1}}$).

The present study was motivated by the wish to test the hypothesis that an even larger model domain would allow the MCS to be simulated. We aimed to (i) produce a reasonable hindcast of this event by making the model domain large enough to cover all stages of the event and (ii) investigate what aspects of the model domain (e.g. size, position) were important to successfully simulate the MCS. For regional climate simulations, the sensitivity to the size and position of the domain chosen is well known (e.g. Miguez-Macho et al., 2004). In the study by Miguez-Macho et al. (2004), the center of the grid was successively moved 17° to the west, 10° to the east, 7° to the north, and 10° to the south. These large changes led to a distortion of the large-scale circulation by interaction of the modeled flow with the lateral boundaries

of the nested domain which sometimes had a large effect on the precipitation results. Seth and Giorgi (1998) demonstrated that the domain of a regional climate model must be carefully selected for its specific application. In particular, domains much larger than the area of interest appear to be needed for studies of sensitivity to internal forcings, as the interactions between boundary conditions and internal model forcings played an important role.

Beside the influence of different domain sizes, the approach of shifting the model domain boundaries (and keeping the number of grid points constant) has been rarely used for short-range convection-resolving numerical weather prediction. The only study, to the authors' knowledge, was conducted by Henneberg et al. (2018) for examining soil moisture influences on convective precipitation over northern Germany. Perturbations were introduced by shifting the domain boundaries by ten to 30 grid points north and eastwards. Their results have shown that by shifting the model domain, an estimate of the uncertainty of the model results can be calculated and a sufficiently large model spread can be achieved. A somewhat similar technique was used by Schlüter and Schädler (2010) to study the impact of small changes in the synoptic situations on extreme precipitation events. They shifted the large-scale atmospheric fields to north, south, east, and west with respect to the orography by about 28 and 56 km and found that the modeled precipitation can be quite sensitive to small changes of the synoptic situation with changes on the order of 20% for the maximum daily precipitation.

By investigating what aspects of the model domain (size, position) were important to successfully simulate the MCS, we found that small movements of the model domain, e.g. 50 km farther west or slightly increasing all dimensions of the model domain, could separate successful simulations of the MCS from complete failures. We then systematically reduced the size of our model domain changes, eventually converging on a single grid-point (2.8 km) shift in location while keeping the domain size the same. Even with these tiny changes, the representation of the MCS ranges from good to poor.

It is these final simulations, which result from a single grid-point shift in the model domain, that are discussed in this paper. We evaluate what the differences between the simulations were and what the origin of these differences was. This gives us further insight into the important physical processes for this event, and helps understand why it was so difficult to predict in the operational forecast models.

## 2  Synoptic situation and observed precipitation

To describe the synoptic situation of the event, we briefly summarize the analysis from Barthlott et al. (2017). For more details, we refer to that paper and to the synoptic analysis performed by Mathias et al. (2017). The synoptic situation on 9

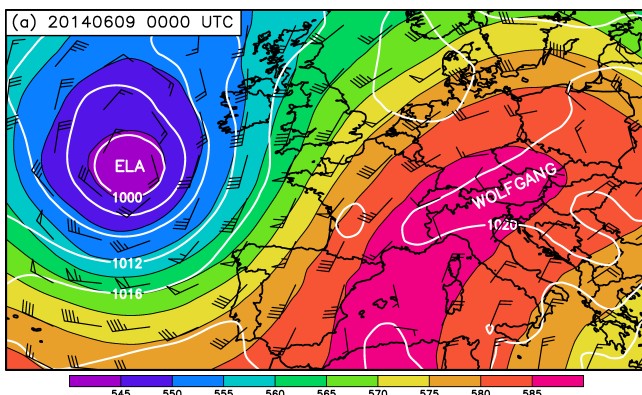

**Figure 1.** Global Forecast System analyses at 9 June 00:00 UTC showing 500 hPa geopotential height (gpdm, shading), sea-level pressure (hPa, white contours), and 500 hPa wind barbs.

June 2014 was characterized by a trough stretching across the northern Atlantic Ocean southwards almost to the Canary Islands and an extensive ridge covering central northern Africa, the western Mediterranean Sea, and central Europe (Fig. 1). At the surface, there was a low pressure system named "Ela" corresponding to the upper-level trough. The high pressure system over the continent ("Wolfgang") dominated the region between the Alps, Poland, and the Black Sea. This configuration was already present on the day before and had progressed only slowly eastward. During the period of 8–10 June 2014, the temperature contrast over Western Europe intensified. Cool Atlantic air masses were present at the eastern edge of the low pressure system, while moist and very warm air of subtropical origin was carried northeastward by the strong upper-level south-westerly flow.

Intense thunderstorms developed in northwestern France and the Benelux countries during the night and in the morning hours of 9 June 2014 and also later in the day due to diurnal surface heating. In the evening, an elongated area of convective storms extended from eastern Spain across western and northern France all the way to Benelux (i.e. Belgium, the Netherlands, and Luxembourg) and northwestern Germany. An intense MCS reached its mature phase in the evening over Benelux and western Germany, which is in the focus of this study. The analysis of satellite pictures in Fig. 2 reveals that the system originated over the Bay of Biscay in the morning of 9 June. The temporal evolution was characterized by several cycles of intensification and decay. For example at 16:00 UTC (Fig. 2e), an intensification at the northeastern edge of the system took place which lead to the large MCS over Germany in the evening with overshooting tops and signs of gravity waves (Fig. 2f).

## 3  Method

### 3.1  COSMO model

All simulations were performed with version 5.3 of the numerical weather prediction model COSMO (COnsortium for Small-scale MOdeling, Schättler et al., 2019). The COSMO model is a nonhydrostatic limited-area atmospheric prediction model initially developed by the Deutscher Wetterdienst (DWD, German Weather Service) which is operationally used by several weather services in Europe. It is based on the fully compressible primitive equations integrated with a two-time level Runge-Kutta method (Wicker and Skamarock, 2002). As previous simulations of Barthlott et al. (2017) showed little sensitivity of the results to model grid spacing, we performed all simulations with 2.8 km horizontal grid spacing and 50 terrain-following vertical levels. This corresponds to the operational used setup at the DWD at the time of the event. For consistency with previous simulations of this case, the changes suggested by Barrett et al. (2019) to minimize timestep-dependent results from the microphysics parameterization were not included. The model uses an Arakawa C-grid for horizontal differencing on a rotated latitude/longitude grid. Initial and boundary conditions come from the ECMWF's Integrated Forecasting System (IFS) analyses with a resolution of 0.125 °. All simulations are initialized at 00:00 UTC on 9 June 2014 with an integration time of 36 h. The time step is set to 25 s. Deep convection is resolved explicitly and a modified Tiedtke-scheme (Tiedtke, 1989) is used to parameterize shallow convection (as did the operational deterministic and ensemble prediction system at that time). A 1D turbulence parameterization based on the prognostic equation for the turbulent kinetic energy after Mellor and Yamada (1974) is applied. No latent heat nudging or other data assimilation technique is used. Instead of the operationally used single-moment microphysics scheme, we use the double-moment scheme of Seifert and Beheng (2006) assuming continental concentrations of cloud condensation nuclei ($N_{CN}$ = 1700 cm$^{-3}$). In our configuration, the CCN concentration remains constant and is not varied as, for example, in the study of Barthlott and Hoose (2018) investigating aerosol effects on clouds and precipitation in central Europe.

### 3.2  Model domain choices

At first, we conducted a reference run (REF) with a model domain containing 600×650 grid points, which corresponds to an area of about 1668 km×1807 km. To be able to simulate the entire life cycle of the convective system, the domain covers France, Benelux, Germany, and the Alps with parts of the neighboring countries (Fig. 3). The sensitivity of the model results to domain shifting is assessed by conducting simulations where the model domain is shifted by one grid

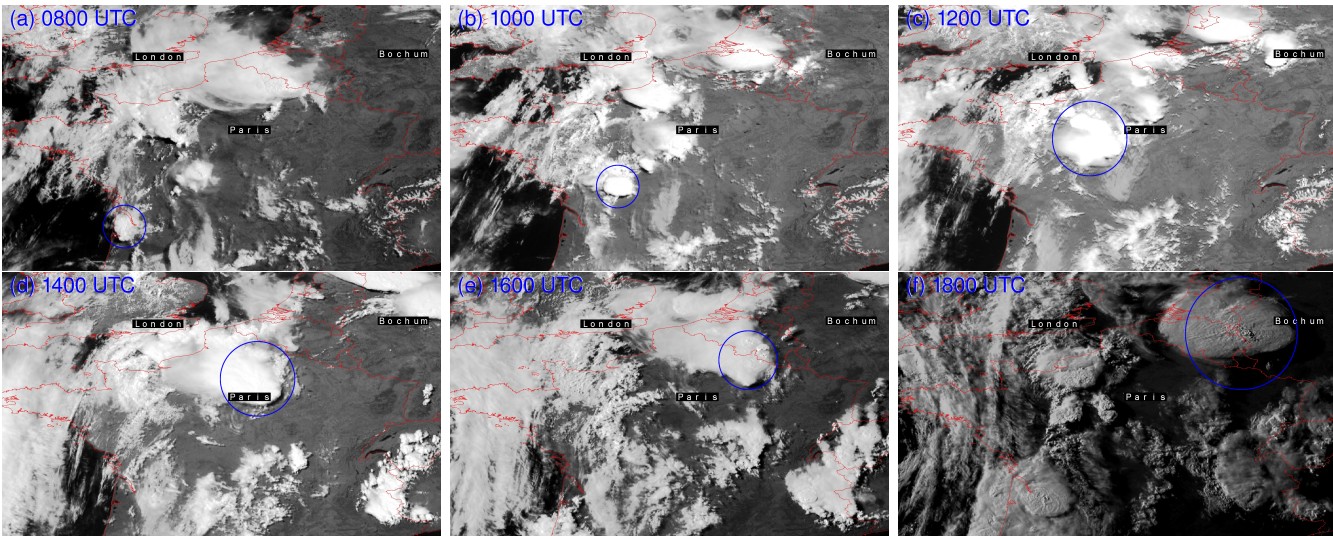

**Figure 2.** Meteosat visible satellite pictures of northern France and western Germany from 08:00–18:00 UTC (raw data courtesy of EU-METSAT). The convective system investigated here is marked by the blue circles.

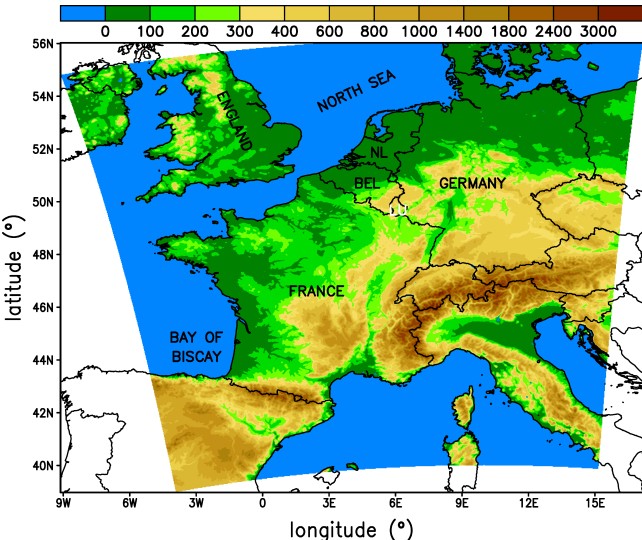

**Figure 3.** COSMO simulation domain and model orography (in m asl) of the reference run.

point in eight different directions (Table 1). All other model settings remained unchanged.

## 4 Results

### 4.1 Reference run

Here we compare our reference simulation to radar-derived precipitation from the precipitation analysis algorithm RADOLAN (Radar Online Adjustment), which combines weather radar data with hourly surface precipitation

**Table 1.** Overview of the numerical simulations.

| name | domain shifting |
|------|-----------------|
| REF  | none |
| N    | 1 grid point towards the North |
| NE   | 1 grid point towards the North-East |
| E    | 1 grid point towards the East |
| SE   | 1 grid point towards the South-East |
| S    | 1 grid point towards the South |
| SW   | 1 grid point towards the South-West |
| W    | 1 grid point towards the West |
| NW   | 1 grid point towards the North-West |

observations of about 1300 automated rain gauges to get quality-controlled, high-resolution (1 km) quantitative precipitation estimates. In the reference run, simulated precipitation on the evening of 9 June occurs over Benelux and northern Germany (Fig. 4b). The area covered by precipitation generally agrees well with that from radar observations (Fig. 4a). However, the simulated precipitation is slightly too far north. In areas near Cologne, Frankfurt, and south of Karlsruhe, the model produces less precipitation than observed and some single convective cells are not simulated. In contrast, precipitation covers more of the English Channel northern Netherlands and Belgium than observed. As far as the total precipitation amounts are concerned, the COSMO model produces similar values to those observed with slightly lower maximum values. However, radar is not an instrument measuring precipitation in a quantitative sense (see e.g., Rossa et al., 2005) and differences in the amount do not necessarily indicate a poor performance of the model. Unfortunately, this radar composite also suffers from missing

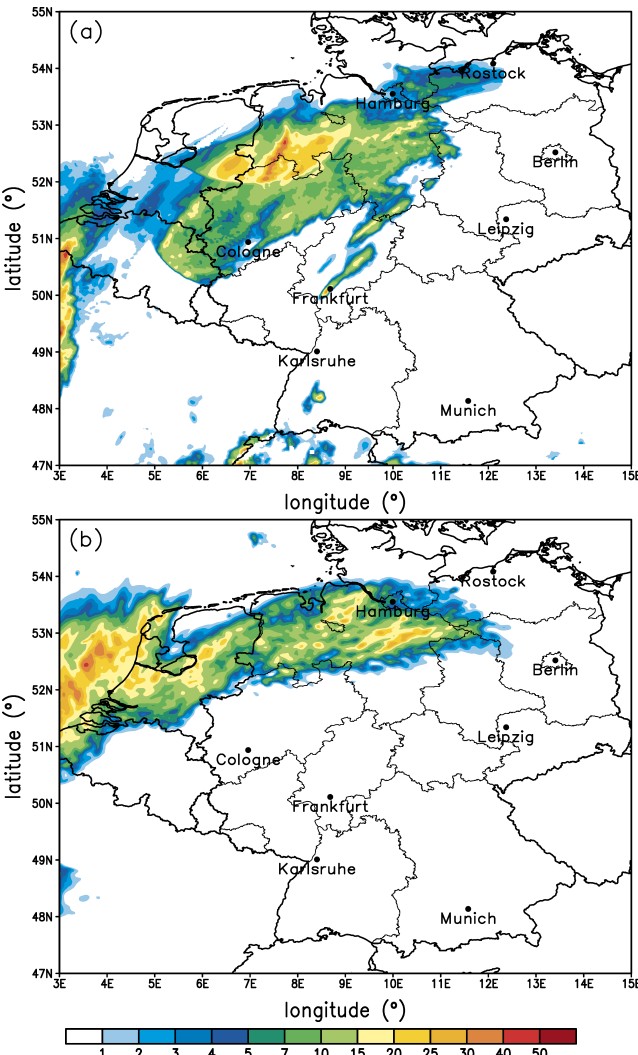

**Figure 4.** Radar-derived (a) and simulated (b) accumulated precipitation in mm on 9 June 2014 (17:00–24:00 UTC).

data at some locations (e.g. over Belgium south-west from Cologne) and also different calibrations or Z-R-relationships (obvious from the strong precipitation gradient about 100 km north of Cologne).

The temporal evolution of the convective system from both radar-derived and simulated 30-min precipitation rates is presented in Fig. 5. Both systems follow a very similar track. We observe the following two main differences: (i) the model simulates the convective system too far to the North and (ii) the simulated MCS moves faster towards the East. These differences are similar to the simulations of Mathias et al. (2017). Moreover, the observed area covered with rain is larger than simulated. Given the overall good agreement in precipitation location and timing with reasonable accumulations, we conclude that the reference run serves as a good basis for our sensitivity studies.

## 4.2 Sensitivity to domain choice

The 24 h accumulated precipitation for the REF run and all shifted model runs is shown in Fig. 6. All model realizations show convective systems initiated near the Bay of Biscay in southwestern France which later move towards the northeast. However, these systems are not related to the life cycle of the MCS that forms later over Germany and are not important for this study. The system that later became the MCS started as several smaller convective showers near the city of Nantes in the morning hours (starting around 06:00 UTC). Showers were initiated over the sea by a combination of large-scale forcing (determined by Q-vector divergence) and low-level wind convergence (not shown). The subjectively determined track of the system in the REF run is marked by the red lines in all model runs. This first convection initiation is displaced to the North compared to the satellite observations (Fig. 2), which was nearer Bordeaux, and explains the northward displacement of the MCS track over Germany later in the evening. In addition to the REF run (Fig. 6e), the runs NW, N, SW S, and, to a lesser extent also run E, successfully simulate convective precipitation over northern Germany. In run E (Fig. 6f), the area with precipitation is too far north and the system decays too early, west of Hamburg. The other successful model runs differ slightly from REF in the maximum rain amounts and horizontal extent of precipitation on the ground. Nevertheless, the results of those runs is rather similar with respect to 24-h accumulated precipitation. From these accumulations alone, the runs N, NW, or SW seem to be better suited as reference simulation due to the larger precipitation amounts. However, the analysis of the temporal evolution (not shown) reveals that the reference run is closest to observations when both rain distribution and temporal development are considered.

The runs without any deep convection over northern Germany are the runs NE, W, and SE. Except for some weak and isolated showers north of Cologne in the W run, there is no precipitation simulated in the region of interest. Given that the model domain was shifted by only one grid point, this pronounced difference in the simulation results is unexpected. All of these unsuccessful runs simulate more precipitation over the English Channel and the coastal regions of the Netherlands than the REF run. Additionally, there is no systematic response of the model to domain shifting in any direction, e.g. there is no systematic decrease of precipitation when shifting the domain from North to South or East to West and the three unsuccessful simulations are not adjacent to one another.

## 4.3 Differences in initial and boundary conditions

As we shifted the model domain only by one grid point towards the eight possible directions (referred to as Queen's case in spatial statistics), we expect only small differences in the initial and boundary conditions. This is justified by the

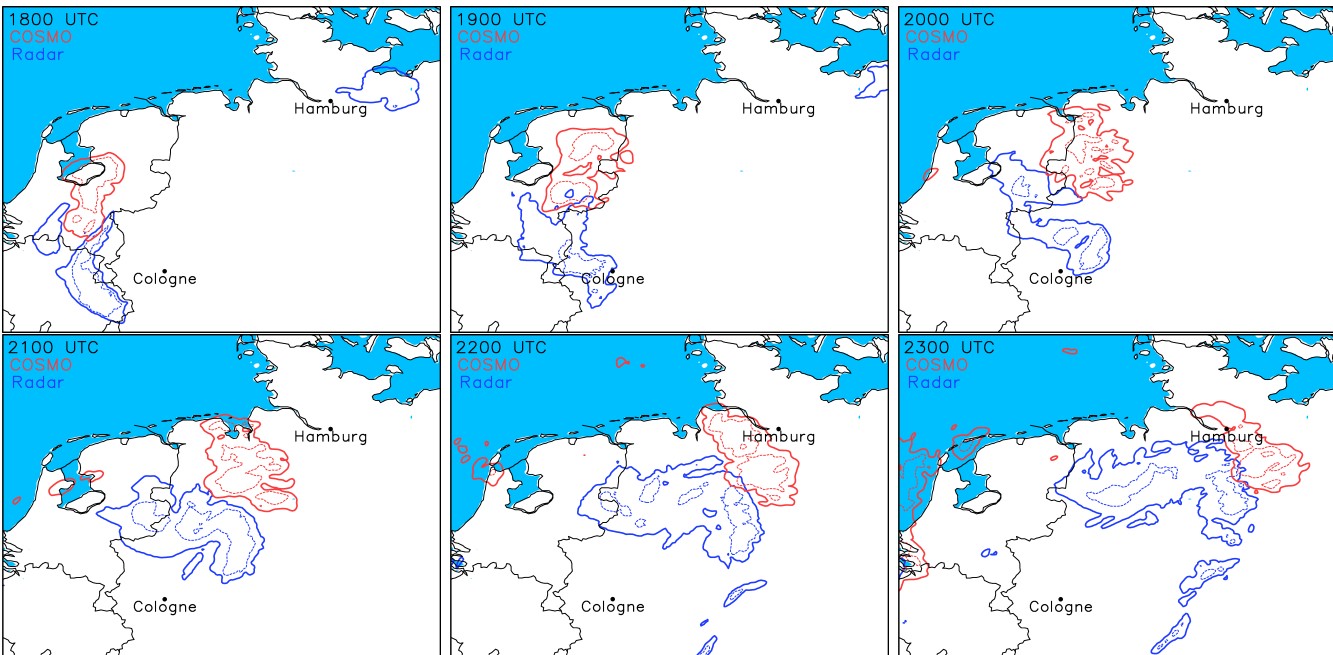

**Figure 5.** Radar-derived (blue contours) and simulated (red contours) 30-min precipitation of 1 mm (solid) and 5 mm (dashed) on 9 June 2014.

difference in horizontal resolution of the initial data and the one used for the COSMO simulations. The spatial resolution of the IFS analyses used in this study is approximately 13 km. As the COSMO simulations are run with 2.8 km grid spacing, many of the grid points used in the preprocessor are the same if they are shifted by $\Delta x = 2.8$ km. This circumstance is illustrated in Fig. 7 in which the grid boxes of the input data and the COSMO grid of the southwest corner are displayed. Only for parts of the model boundary does the domain shifting of the high-resolution grid also imply a different grid point used for interpolation in the preprocessor of our model. Moreover, even when analyzing only the IFS input data, we do not see large point-to-point gradients in any meteorological fields near the boundary of the nested COSMO simulations (not shown).

However, small differences are present and assessed quantitatively by domain-averaged meteorological variables at initialization time (Table 2). Neither the surface fields (2 m temperature and specific humidity), nor the vertically-integrated variables (convective available potential energy CAPE, convective inhibition CIN, 2.5–5-km averaged relative humidity RH, liquid water path LWP, ice water path IWP, and deep layer shear DLS) does the model simulate any large differences in our ensemble of simulations. For example, the 2 m temperature differs by a maximum of 0.1°C between individual model runs. It is also of interest to investigate if the lateral boundaries (updated every 6 h) show any differences when the model domain is shifted. We therefore calculated averaged profiles for each of the four model

**Table 2.** Domain averaged 2 m temperature (T in °C), 2 m specific humidity (QV in g kg$^{-1}$), convective available potential energy (CAPE in J kg$^{-1}$), convective inhibition (CIN in J kg$^{-1}$), 2.5–5 km averaged relative humidity (RH in %), liquid water path (LWP in g m$^{-2}$), ice water path (IWP in g m$^{-2}$), and deep layer shear (DLS in m s$^{-2}$) at initialization time.

| run | T | QV | CAPE | CIN | RH | LWP | IWP | DLS |
|-----|------|-------|-------|-------|------|------|------|------|
| NW  | 18.5 | 10.35 | 215.0 | 297.7 | 44.8 | 0.32 | 2.72 | 14.1 |
| N   | 18.5 | 10.35 | 214.7 | 297.6 | 44.7 | 0.32 | 2.74 | 14.1 |
| NE  | 18.5 | 10.36 | 214.8 | 297.5 | 44.7 | 0.32 | 2.76 | 14.1 |
| W   | 18.5 | 10.36 | 215.0 | 298.1 | 44.7 | 0.32 | 2.72 | 14.1 |
| REF | 18.5 | 10.36 | 214.8 | 298.0 | 44.7 | 0.32 | 2.74 | 14.1 |
| E   | 18.5 | 10.36 | 214.9 | 297.9 | 44.7 | 0.32 | 2.76 | 14.1 |
| SW  | 18.5 | 10.37 | 215.1 | 298.6 | 44.7 | 0.32 | 2.72 | 14.1 |
| S   | 18.5 | 10.37 | 214.9 | 298.4 | 44.7 | 0.32 | 2.73 | 14.0 |
| SE  | 18.6 | 10.37 | 214.9 | 298.3 | 44.6 | 0.32 | 2.76 | 14.0 |

boundaries for temperature, specific humidity and both horizontal wind components. The analysis of probability distributions (not shown) reveals that the range of simulated values is identical for all variables and only minor differences in the frequency of occurrence exist. Furthermore, averaged values of those profiles are compared for every lateral boundary condition file (not shown). The maximum difference of the sensitivity runs to the REF runs is 0.02 K for temperature, 0.01 g kg$^{-1}$ for specific humidity, and 0.1 m s$^{-1}$ for the wind components. We therefore conclude that all differences

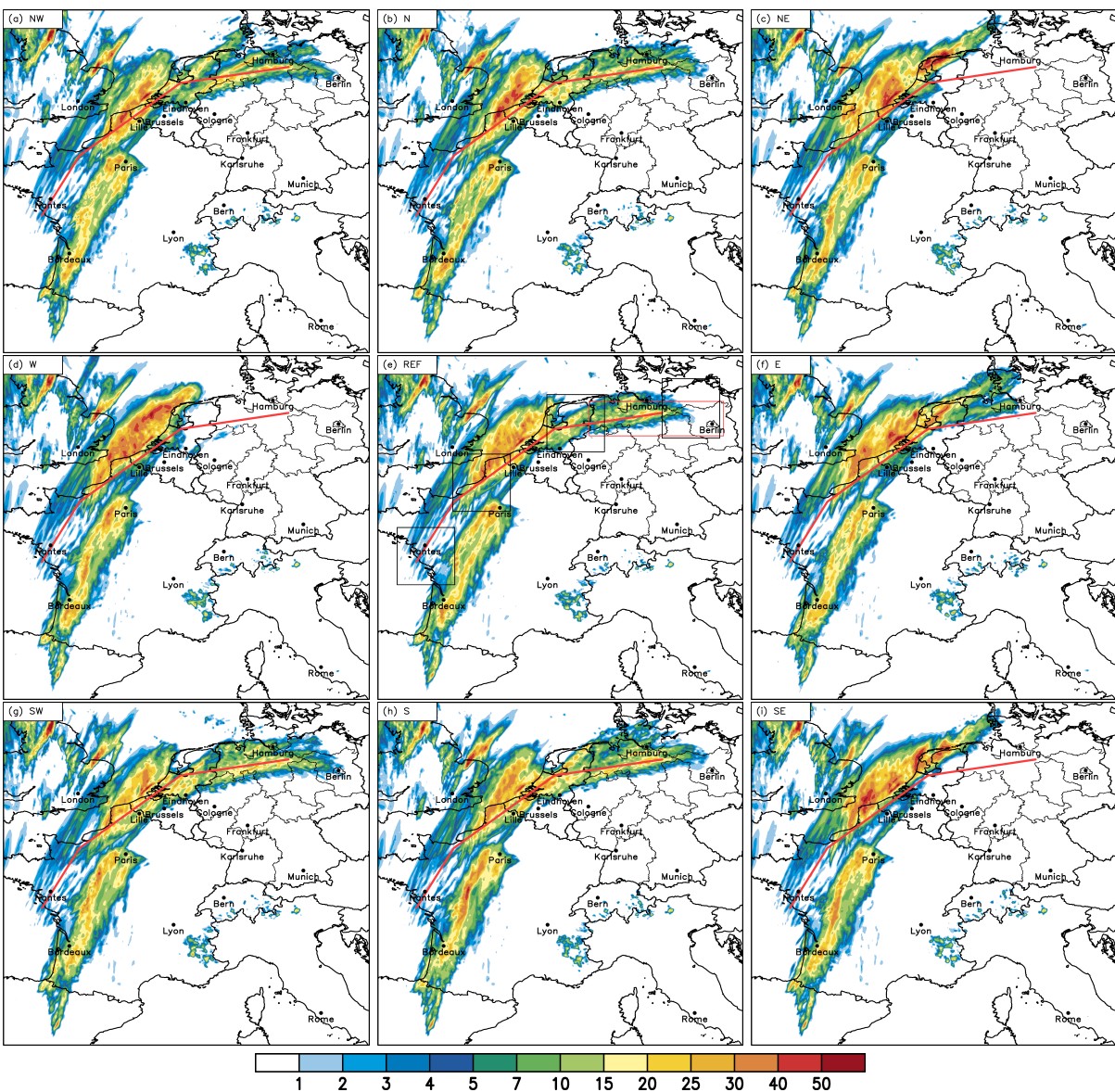

**Figure 6.** 24-h precipitation (00:00–24:00 UTC on 9 June) amount in mm. The red line indicates the approximated storm track of the REF run. The black boxes in (e) indicate areas for averaging convection-related variables along the storm path for 06:00, 12:00, 18:00 and 24:00 UTC (from left to right). The red box in (e) depicts the area for calculating precipitation over Germany for the ensemble sensitivity analysis presented in section 4.5.

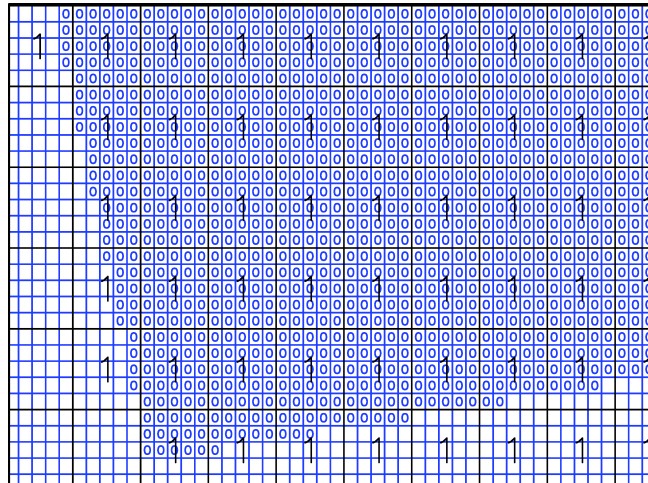

**Figure 7.** Southwest corner of the simulation domain with illustration of IFS grid (black) and COSMO grid (blue). Numbers of 1 indicate IFS grid points whereas 0 indicates COSMO grid points.

in the initial and boundary conditions of the domain-shifted model runs are small.

### 4.4    Convection-related parameters

The general preconditions for the initiation of deep moist convection are (i) conditional instability, (ii) a sufficient amount of humidity in the lower and middle troposphere to form clouds, and (iii) a trigger process to bring air parcels to their level of free convection (e.g., Doswell III, 1987; Bennett et al., 2006). Trigger processes are e.g., the reaching of the convective temperature, lifting by convergence zones (e.g., Crook and Klemp, 2000), or terrain-induced ascent (Kirshbaum et al., 2018). The organization and further life cycle is then affected by the vertical wind shear, CAPE, and relative humidity. To assess the state of the atmosphere in the vicinity of the MCS affecting northern Germany, we calculated several convection-related variables averaged over a rectangular box surrounding the convective system. The box has a size of $3° \times 2.5°$ and follows the storm along the path depicted in Fig. 6. The box has been positioned in such a way that the convection is not centered in the domain, but rather on the western edge to better capture the (preconvective) environment into which the storm is moving (see Fig. 6e for the location of this box at four selected times).

Figure 8 presents time series of some of these parameters during the life cycle of the convective storm. The brightly colored lines represent the successful simulations, black is the reference simulations and gray and blue colored lines represent the unsuccessful simulations. The precipitation rate of the REF run is gradually increasing until 14:30 UTC (Fig. 8a) and remains more or less constant until 17:00 UTC. The precipitation rate is strongly increasing until 18:30 UTC followed by a slight reduction in intensity before the max-

imum is reached between 21:00 UTC and 22:00 UTC. After 22:00 UTC, the precipitation rate decreases and the convective system slowly dissipates. The other successful runs (NW, N, SW, S) show larger precipitation rates and an earlier increase already from 12:00 UTC. As already mentioned earlier, these runs agree less well with observations in terms of precipitation location and timing than the REF run. The runs without an MCS over northern Germany (W, NE, SE) simulate similar precipitation amounts to the other runs until 11:00 UTC, but then rain gradually stops. Only run W simulates longer lasting precipitation until 14:00 UTC and a minor peak from a short-lived cell east of Eindhoven at 17:30 UTC. The 0-6 km deep layer shear (Fig. 8b) is similar in all model runs with values of $27–30\,\mathrm{m\,s^{-1}}$. Such high values indicate suitable conditions for highly-organized convection to develop in all runs because the precipitation and outflow become separated from the low-level updraft. Before the storms form there is almost no difference between the speed or direction of the wind shear in any of the simulations. There is plenty of moisture available for convection, and both the mid-level relative humidity (Fig. 8c) and precipitable water (Fig. 8d) show large values that increase as the storm environment moves further East later in the day. The simulations are again all very similar. The maxima in relative humidity are reached at 19:30 UTC which corresponds to the period with highest rain intensities. As the differences in relative humidity between the individual model runs are very small (2–4%), we determine that evaporation or entrainment processes are not responsible for the different model results. Moreover, between 14:00–22:00 UTC, the mid-level relative humidity is always higher than 60% which suggests that the role of entrainment of drier environmental air is probably only small. The same applies for the precipitable water for which all model realizations lie close together until 14:00 UTC (Fig. 8d). At later times, the precipitable water is affected by different rain formation and evaporation processes.

Additionally all simulations show substantial conditional instability, especially later in the day. Before 11:00 UTC, all simulations have similar amounts of CAPE (Fig. 8e). Later on, in simulations with larger precipitation totals, more CAPE has been consumed. This leaves the runs without an MCS over Germany with the highest CAPE values in the early evening ($2500\text{-}3300\,\mathrm{J\,kg^{-1}}$). For convection initiation or development, CAPE alone is not a suitable parameter. We therefore calculated the fraction of grid points, for which CAPE is larger than $600\,\mathrm{J\,kg^{-1}}$ and CIN is lower than $5\,\mathrm{J\,kg^{-1}}$ (Fig. 8f). Here there is a large contrast in the number of grid-points where convection is expected at 13:00 UTC between the successful (around 10%) and unsuccessful (around 5%) runs. However, the reference run and the unsuccessful runs show rather similar curves. The W run reveals a somewhat lower maximum and a quick decrease afterwards. The secondary maximum occurring at 17:00 UTC corresponds to the aforementioned isolated cell initiated near

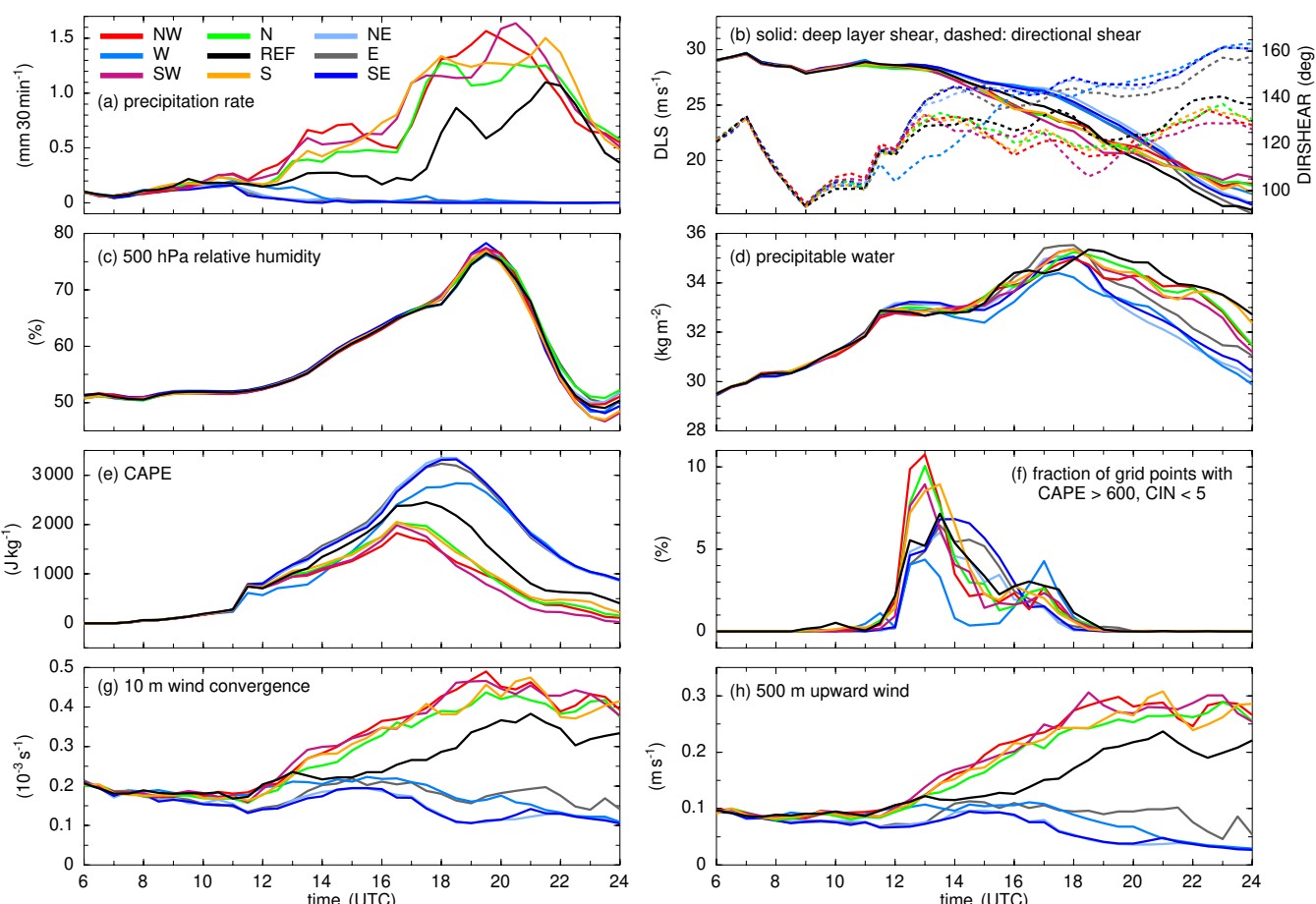

**Figure 8.** Time series of convection-related parameters averaged over a $3° \times 2.5°$ box surrounding and following the convective system. Shear parameters are based on surface and 6-km altitude winds.

Eindhoven. The fraction of grid points fulfilling that criterion is primarily dominated by the existence of CAPE as domain-averaged CIN is very similar in all model runs until 12:00 UTC (not shown).

Low-level wind convergence (Figs. 8g) is one mechanism for producing lift that leads to convection. The time series of convergence values are very similar to the upward vertical motion in the boundary layer (Figs. 8h) which indicates that the lift is primarily produced by convergence, mostly along convective outflow boundaries. Convergence early in the day can not be solely attributed to surface inhomogeneities or terrain features, because small amounts of rain are already simulated in the morning hours leading to wind convergence at outflow boundaries. Between 08:00–11:00 UTC, the convergence of the unsuccessful runs (NE, SE) is slightly weaker despite similar precipitation rates (Fig. 8a). However, after 11:00 UTC there is a clear split between the successful and unsuccessful simulations, with increased convergence and upward wind velocities in the successful simulations. Of the unsuccessful runs, only W exhibits similar convergence

strength and lifting in the boundary-layer as the successful model runs and only until 13:00 UTC.

While this analysis does not completely separate the simulations into successful and unsuccessful subsets, there is information that helps explain the chance of producing an MCS. Clearly increased low-level convergence before 12:00 UTC is a good predictor of the later MCS, as is a large increase in the number of grid points with high CAPE but low CIN at 13:00 UTC. The lower CAPE and reduced deep-layer shear in the successful runs after 15:00 UTC are evidence of the storm modifying its own environment rather than a useful predictor of the MCS.

## 4.5 Ensemble sensitivity analysis

The above analysis has shown that whether the simulation is successful or not can be quantified based on small differences in the environment close to the developing convective cell. However, from this analysis we cannot tell what causes these changes in the preconvective environment. Here we use ensemble sensitivity analysis to help identify the origin of these differences. This analysis determines geographical ar-

eas, model variables and times that are correlated with a successful simulation. Although the correlations do not provide evidence of a causal relationship they do provide a starting point for understanding the diverging simulations, as shown in the below analysis and also by Barrett et al. (2015), Bednarczyk and Ancell (2015), Hill et al. (2016), Torn et al. (2017).

Following the above papers, we define the ensemble sensitivity $S$ as

$$S = m \, a \, \sigma_x = \frac{\mathrm{cov}(y,x)}{\mathrm{var}(x)} \, a \, \sigma_x \qquad (1)$$

where $m$ is the regression coefficient between the test variable $x$ and the response function $y$ as calculated at each grid point. A scaling factor $a$ is used to de-emphasize noise in the analysis based on the correlation coefficient $r$; $a = 0$ where $r^2 < 0.4$ and $a = 1$ otherwise. Finally the sensitivity is scaled by the ensemble standard deviation of the test variable $\sigma_x$, calculated individually at each grid point, to normalize the calculated sensitivity; this enables comparison of sensitivities to different model variables.

We attempted to use ensemble sensitivity analysis on numerous model variables at surface, 850, 500 and 250 hPa levels. However, because the model initial states are nearly identical, the ensemble sensitivity analysis is unable to identify any relationships between convection intensity and typical large-scale drivers of convection before the convection develops. Only after the convection develops can signals be seen in the e.g. upper-level pressure, wind and divergence fields as they are directly modified by the convection. Therefore the analysis below focusses on CAPE, precipitation rate and vertical velocity. These variables are those shown in Fig. 8abd also discussed in more detail in section 4.6 and Fig. 10.

The sensitivity is calculated using the mean precipitation over northern Germany between 12:00–23:00 UTC (box bounded by 7 W, 14 W, 52 N, 53.5 N; red box in Fig. 6e), which is 4.4–6.7 mm in the successful ensemble members 1.0 mm in the E-shifted member and less than 0.1 mm otherwise. The resulting sensitivity $S$ has units of mm of precipitation per standard deviation change in $x$ (here CAPE, precipitation rate or vertical velocity $W$). Hence the ensemble sensitivity is interpreted as the change to precipitation for a one standard deviation increase in variable $x$ at that grid point.

The ensemble sensitivity analysis (Fig. 9) shows a correlation between increased precipitation over northern Germany with lower CAPE over northern France earlier in the day (11:00-13:00 UTC; left column of Fig. 9). Precipitation located at the northern end of this region of low CAPE is also correlated with successful simulation of the MCS (middle column of Fig. 9). There is also a weak signal of sensitivity to weaker vertical velocities over the region of sensitivity to low CAPE (right column of Fig. 9), with stronger vertical velocities located to the east of the low CAPE re-

gion (12:00–13:00 UTC). However, much of the sensitivity to vertical velocity appears to be noise resulting from small changes in location of updrafts in the different simulations. These signals should be interpreted as a relationship between increased precipitation across northern Germany in ensemble members with a strong surface cold pool over northern France earlier in the day (at locations marked by the negative sensitivity to CAPE). The signal is first evident in sensitivity to CAPE at 11:00 UTC. However, the signal is much weaker at 09:00 UTC as the cold pool is yet to develop. A convective system exists near 47.8 N, 0.75 W in all ensemble members at 09:00 UTC (not shown) but the decisive factor regarding later MCS development seems to be the exact location at this time, with a location farther to the east favoring MCS development. This does not show up well in the ensemble sensitivity analysis because the disparate locations of the cells mean that values at individual grid-points are not correlated with later success.

The disturbance at 09:00 UTC can be tracked farther back to the western french coast, around 150 km south of Nantes (46 N, 1 W) already at 04:00 UTC (not shown). However, neither the ensemble sensitivity analysis nor more detailed investigation into the convective disturbances at this time showed any systematic structure of the convective cells that were decisive in the successful simulation of the MCS later in the day. The important aspect appears to be that by 09:00 UTC that a line of convection begins to form on the outflow of this convection, and that changing the position of that cell by only around 10 km determines whether the convective cell evolves into the MCS which later affects Germany, or not.

The ensemble sensitivity analysis has helped highlight interesting areas in the development timeline of the convective cells. However, due to the disparate locations of the convective cells, the grid-point correlations required for ensemble sensitivity analysis do not help explain how these cells differ in their development. In the next section we evaluate in more detail how the developing convective cells interacted with their environment and what caused the differing convective evolutions.

## 4.6 Simulation result differences

In this section we discuss horizontal cross-sections of convection-related parameters to elucidate the differences (and their possible causes) of the different model results. For the sake of brevity, we only compare the reference run to two unsuccessful runs, namely the ones with the model domain shifted towards the W and the SE. Figure 10 presents a time series of those cross-sections for the region of northwestern France and southern England.

The analysis at 10:00 UTC shows a very similar picture for all simulations at all times, with CAPE increasing to the South and East, wind is westerly over the English Channel, turning to northerly direction over France and there is low

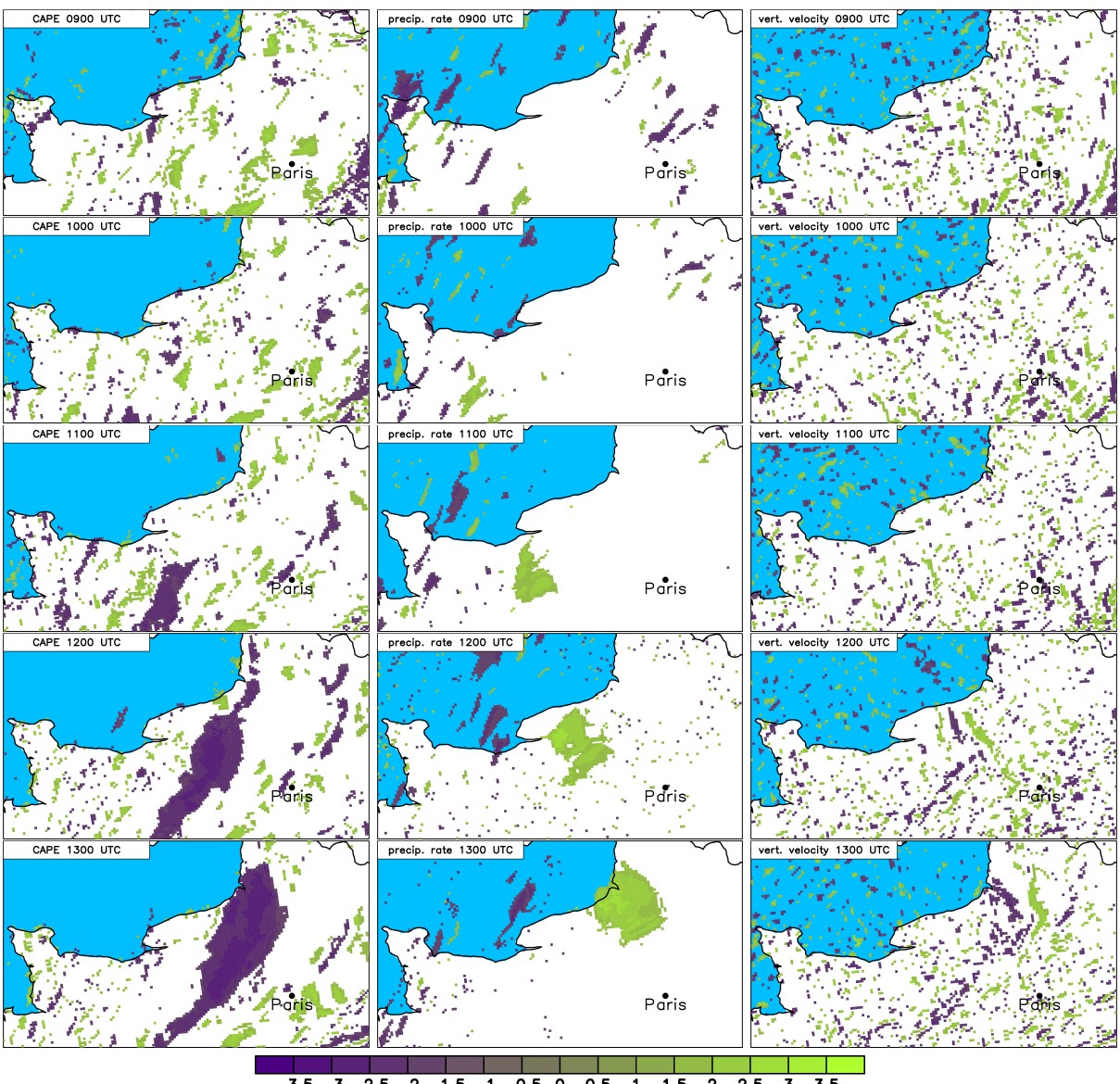

**Figure 9.** Ensemble sensitivity of precipitation over Germany to CAPE (left column) and precipitation rate (middle column) and vertical velocity at 500 m height (right column). The units are mm per standard deviation change in the ensemble, with positive values indicating that ensemble members produced more precipitation over Germany when the CAPE, precipitation rate or vertical velocity in the marked locations was larger in the ensemble. A signal of increased precipitation for lower CAPE values develops throughout the morning, consistent with a large developing cold pool (left column). The precipitation in the successful simulations is at the northern end of this cold pool (middle column); however, the location of the low-level updrafts is too variable in the ensemble to be seen with this analysis (right column).

CIN over France from 11:00 UTC at the latest. The simulations all have weak, disorganized convection over northwestern France and a more isolated cell at the border between France and Belgium. The region with the convective system of interest is marked with a red circle. However, as time develops, only the REF simulation produces a convective system that moves into the high CAPE region to the East and later becomes an MCS over Germany. By 10:30 UTC, CIN is less than 5 J kg$^{-1}$ in the REF and SE run, whereas it is still above that threshold until 11:00 UTC in the W run. The highest rain intensities are also simulated in the REF run. At 11:00 UTC approximately half of the cell of interest in the W run is over the sea, where CAPE is lower and CIN higher than over land. In contrast, the area of convective rain in the REF run is separated from the rain over the sea and precipitation intensity remains high. The precipitating area in the SE run is also separated from the larger rain area over the sea, but the precipitation rate is already weaker than in the REF

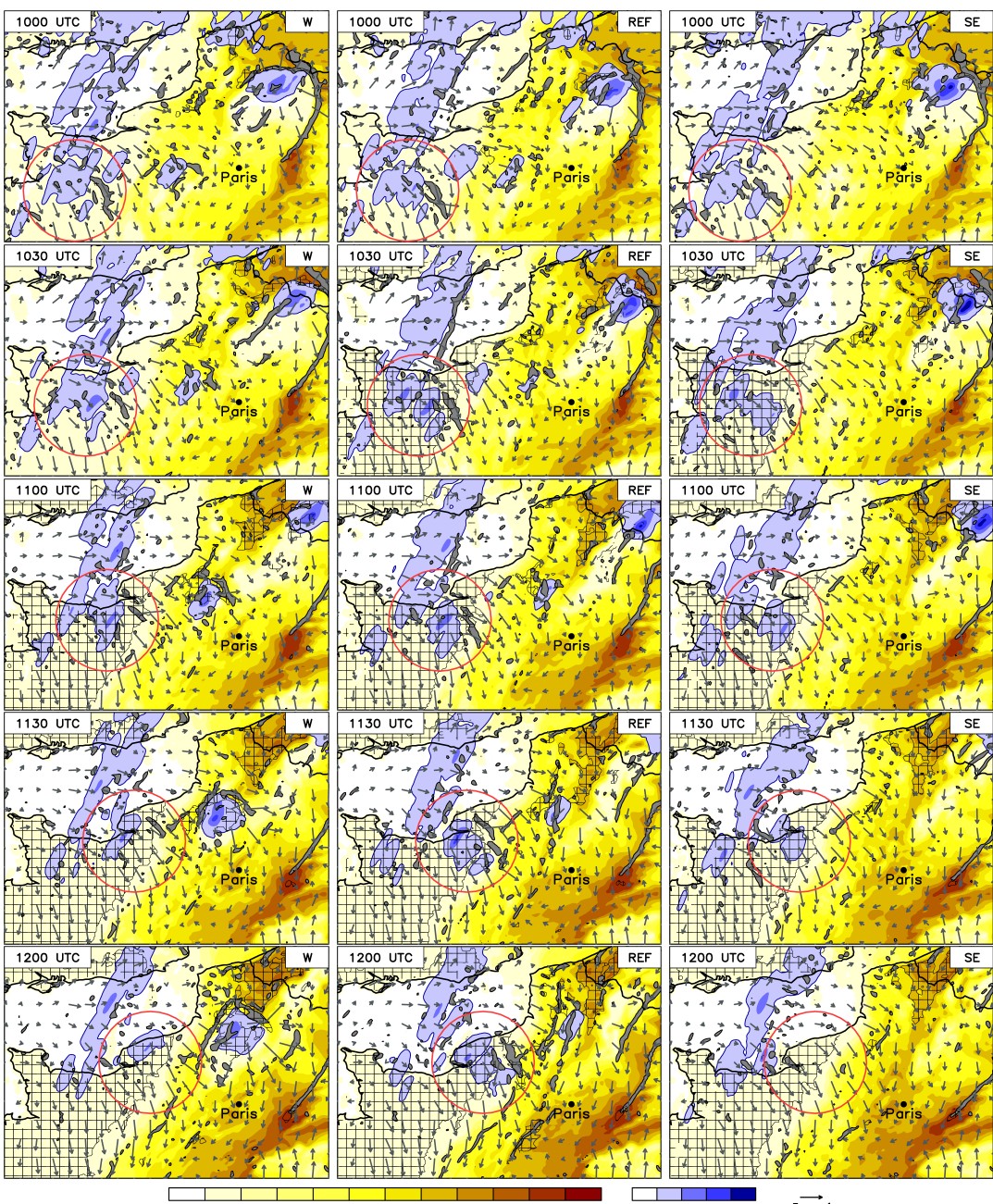

**Figure 10.** Convective available potential energy (color shading, in $\mathrm{J\,kg^{-1}}$, 30-min precipitation rate (blue color shading, in $\mathrm{mm\,(30\,min)^{-1}}$, and 10-m wind field (arrows) between 14:00–18:00 UTC on 9 June. Gray areas indicate low-level wind convergence larger than $0.35 \cdot 10^{-3}$ $\mathrm{m\,s^{-1}}$ and hatched areas represent regions where convective inhibition is smaller than $5\,\mathrm{J\,kg^{-1}}$. Left: domain shifted to W; Middle: reference run; Right: domain shifted to SE. The red circle indicates the region of the convective cell developing into a MCS.

run. At 11:30 UTC, the cell in the W run is weakening and lies almost at the coastline, whereas the cell in the REF run still remains almost entirely over land while moving towards the North-East. The corresponding system in the SE run is also weakening while traveling towards the North-East; approximately half of the cell is now located over the sea. The systems in the W and REF runs both weaken at 12:00 UTC, but the one from the SE run has already decayed. Only the cell in the REF run intensifies again at 12:30 UTC. In the W run, the cell continues to move along the coastline while weakening, until it is completely dissolved at 14:00 UTC. In the REF run, however, the cell stays almost completely over land and intensifies further while moving towards the Netherlands (13:00–14:00 UTC).

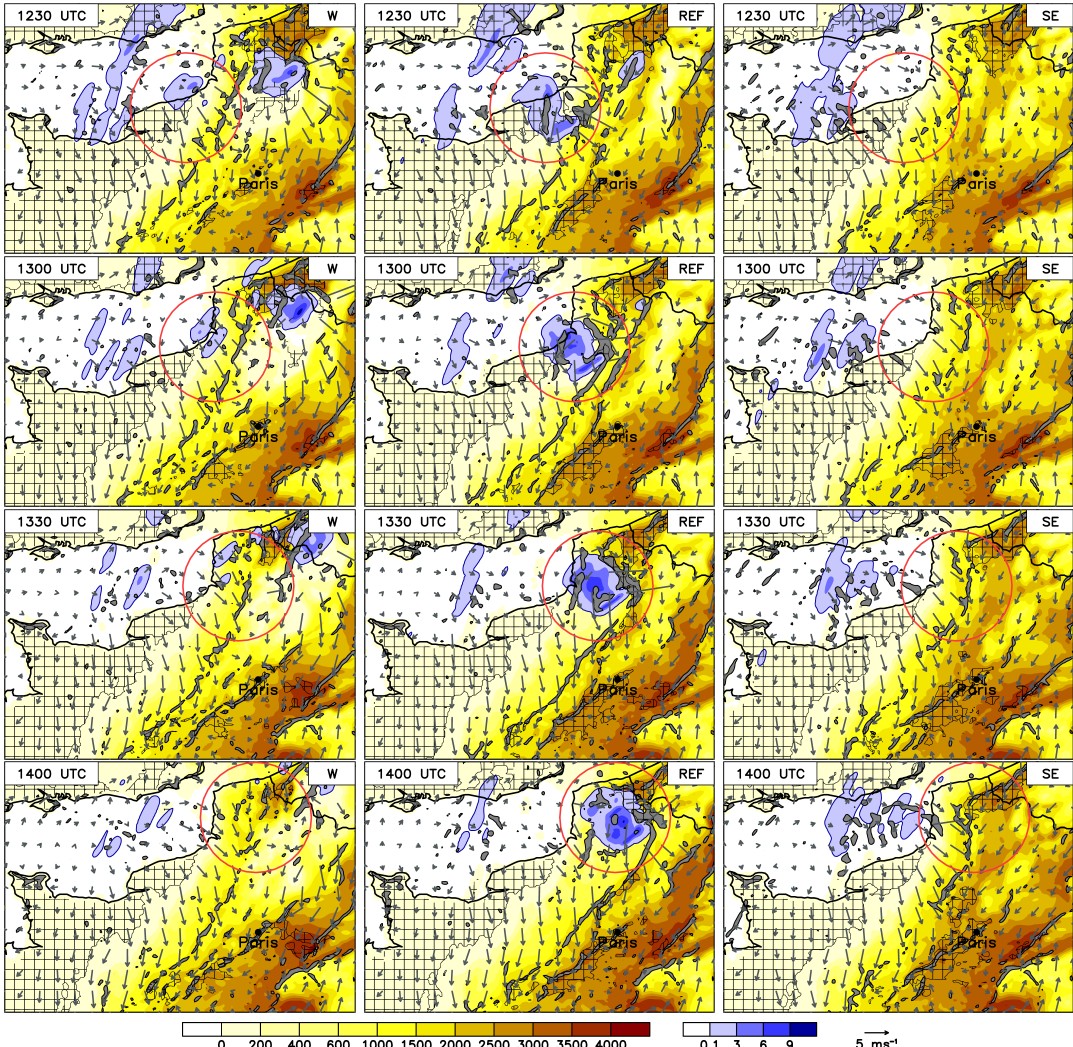

**Figure 10.** Continued.

The sea surface temperatures along the French coast lie around 15°C and are much lower than the land surface temperatures (around 23°C, not shown). This temperature distribution is similar in all model runs for the preconvective environment. The proximity of the cell to the colder sea surface appears to have a decisive influence on the further life cycle of convection. In the REF run and the other successful runs (not shown), the system stays more or less entirely over land between 11:00–12:00 UTC. In these successful simulations, the systems travels further towards Belgium and Germany (rather than over the sea), where it encounters more favorable convective conditions including higher CAPE, which later allows it to evolve into an MCS.

The isolated cell, to the north west of these plots between 10:00-11:00 UTC, does not appear to be important to the decay of the cell of interest. It is located approximately 150 km upstream. The cell is stronger in the W run leading to a slight reduction of CAPE and therefore creating slightly less favorable environmental conditions in the area into which the main cell would later move. However, it appears that the weakening of the main cell occurred independently of the cell upstream and can rather be attributed to the proximity to the colder sea surface.

In addition to this analysis, we further want to point out that the upper-level dynamics are similar in all model runs in the early stage of the convection over France. Hoskins et al. (1978) showed that the traditional form of the quasi-geostrophic omega equation can be rewritten using the Q-vector and that regions of upward (downward) vertical motion are associated with Q-vector convergence (divergence). We calculated the divergence of the Q-vector at 500 hPa and found no noticeable differences between successful and unsuccessful runs, nor are there meaningful differences in geopotential height (not shown). This indicates that the large scale forcing is similar for these model runs and not responsible for the simulation result differences.

## 4.7    Further MCS evolution

Having established a possible explanation for the decay of the precursors of the MCS in the previous section, we now analyze the further evolution of the system into a MCS using the reference simulation (Fig. 11). To the east of the system, the model simulates an east-west oriented region of high low-level equivalent potential temperature in the north-central part of Germany, which corresponds to CAPE values between 3000–4000 $\mathrm{J\,kg^{-1}}$. This CAPE-rich air is advected with easterly winds towards the convective system over the Netherlands. Colliding with the cell's outflow, a strong low-level mass and moisture convergence occurs, which fosters the evolution into a MCS. As already discussed in section 4.4, the 0-6 km deep layer shear shows suitable conditions for highly-organized convection (27–30 $\mathrm{m\,s^{-1}}$). The maximum rain intensities reach locally up to 22 $\mathrm{mm\,(30\,min)^{-1}}$ with a weakly defined bow-like structure of precipitation, typical of storms with an intense rear-inflow jet. In the wake of the MCS, CAPE is almost entirely consumed. From 23:00 UTC onwards, the MCS is decaying while further traveling towards Poland (not shown).

## 5    Comparison of methods to perturb initial and boundary conditions

Many operational forecast centers produce both a high-resolution forecast and an ensemble of lower resolution to provide a measure of uncertainty (Rodwell et al., 2013). There a various ways of generating an ensemble, such as perturbations to the initial conditions and/or boundary conditions (e.g., Montani et al., 2011; Kühnlein et al., 2014), stochastic physical parameterizations (e.g., Buizza et al., 1999; Berner et al., 2017), or ensemble data assimilation such as ensemble Kalman filter (e.g., Dowell et al., 2004; Zhang et al., 2004; Reich et al., 2011). Recent studies by Schneider et al. (2019) and Keil et al. (2019) have also shown that different assumptions for the amount of cloud condensation nuclei could be included in convective-scale ensemble forecasting, but only if the model employs a double-moment microphysics scheme. Because of the fundamental uncertainties of the simulations due to nonlinearities of the model equations, several studies have noted the significant impact of initial boundary conditions (IBC) and lateral boundary conditions (LBC) on the simulation of convective precipitation for some situations (e.g., Hohenegger et al., 2006; Trentmann et al., 2009; Richard et al., 2011; Bouttier and Raynaud, 2018) and that ensemble members with the most accurate initial and boundary conditions are most skilful at predicting the location of convective initiation (Barrett et al., 2015).

One common approach for accounting for uncertainties in the initial and boundary conditions is that perturbations entering the model from the lateral boundaries can be provided by different driving EPS members as it is the case for COSMO-LEPS (Montani et al., 2011) or COSMO-DE-EPS (Gebhardt et al., 2011; Kühnlein et al., 2014). Such perturbations have been shown to play a more and more important role in the behaviour of the limited-area system as the forecast range increases. However, this methology needs an algorithm to select representative members from the driving ensemble (Marsigli et al., 2001). Bouttier and Raynaud (2018) showed that the algorithms used for the member selection have a significant impact on the resulting ensembles and that clustering-based methods outperformed random subsampling. Torn et al. (2006) proposed two classes of methods to populate a boundary condition ensemble. The ensemble of boundary conditions is either provided by an ensemble Kalman filter (EnKF) on a larger domain corresponding to a random draw from the probability distribution function for the forecast (or analysis) on the limited-area domain boundary or by a fixed-covariance perturbation technique. Romine et al. (2014) stated that there remains value in randomly perturbing a deterministic forecast for ensemble lateral boundary conditions. They examined convection-permitting ensemble forecasts by drawing initial conditions from a downscaled ensemble data assimilation system. As the control ensemble was underdispersive, it was supplemented also by perturbed lateral boundary conditions. Lateral boundary conditions for the cycled analysis were generated from perturbed forecasts with the fixed covariance method (Torn et al., 2006). This technique led to a modest improvement in spread and the least degradation in systematic bias.

To further evaluate the large forecast variability of our simulations, we conducted another experiment on the domain of the successful reference run. In this new model run, however, we added small, stochastic perturbations on the boundary conditions, namely random temperature fluctuations with a Gaussian distribution of zero mean and a standard deviation of 0.01 K at all levels. The run does not reproduce the MCS over Germany (not shown) and the rain distribution is similar to the other unsuccessful (e.g. runs NE or SE in Fig. 6). This finding indicates that the simple method of domain shifting can produce a result comparable to stochastic LBC perturbations and could therefore be used to estimate the uncertainty in convection-resolving simulations. In agreement with Henneberg et al. (2018) who used domain shifting by 10 and 30 grid points on a smaller domain combined with soil moisture uncertainties, domain shifting does provide a sufficient large model spread for the case analyzed in this study.

Generally, the approach of domain shifting to generate different IBCs is attractive due to its relative simplicity and practicality of implementation. An advantageous aspect is that IBCs originate from the same driving model, which avoids errors due to inconsistencies that may occur if the initial condition perturbations are applied independently from the boundary condition perturbations (Caron, 2013). Another advantage is the fact that this methodology does not need an algorithm to select representative members if a driving en-

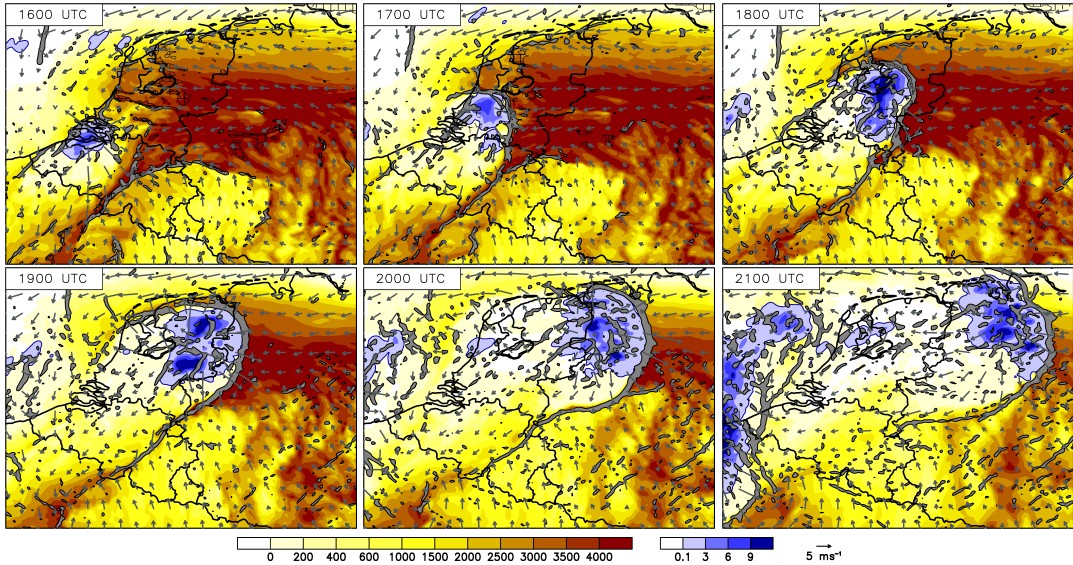

**Figure 11.** Panels as in Fig. 10, showing the development of the MCS over Benelux and Germany between 16:00–21:00 UTC in the REF simulation.

semble is used. However, we want to point out that this study aims at revealing the large forecast variability which can be achieved from domain shifting for this particular case. The applicability of this method needs to be compared with other techniques for this and other cases as well which is left for future work.

## 6   Summary and conclusions

During Pentecost 2014, following a period of hot weather, a mesoscale convective system formed over France and traveled towards Germany in the afternoon of 9 June. A strong southwesterly flow lead to a favorable environment for deep convection due to the advection of warm and moist air. However, the predictability of this event was very low; neither the operational deterministic nor any member of the ensemble prediction system (both convection resolving) captured the event with more than 12 hours lead time (Barthlott et al., 2017).

Hindcasts of this situation were performed with convection-permitting resolution on a large model domain, enabling the simulation of the whole life cycle of the system originating from the western Atlantic coast. The results show that the MCS was reasonably well represented by the COSMO model in this setup. When compared to radar-derived precipitation rates, the MCS was simulated somewhat shifted to the North and the translation speed was slightly higher than observed.

The low predictability of the event was again evident; moving the model domain by just one grid point changed whether the MCS over Germany is successfully simulated or not. The domain was shifted systematically in eight directions (N, NE, E, SE, S, SW, W, NW) by just one grid point and three of these configurations completely failed to simulate deep convection over Germany on that day, while a fourth had some convection but did not capture the organized MCS. This large impact is unexpected when considering the comparatively large computational domain of $1668\,\mathrm{km}\times1807\,\mathrm{km}$.

The evaluation of domain-averaged initial conditions, like low-level temperature, moisture, relative humidity, or wind shear showed only negligible differences. The temporal evolution of convection-related parameters in the vicinity of the storm system also revealed similar conditions in its preconvective environment. The ensemble sensitivity analysis was unable to reveal differences in the upper-level flow between ensemble members, although low-level differences associated with a developing cold pool were identified. An explanation of the large differences in the model results lies in the proximity of the track of the convective system to the north coast of France and the colder temperatures over the sea than the land. The convective system in the successful runs stays more or less entirely over land, allowing it to eventually reach a region favorable for convective organization (with high CAPE, large shear and low CIN), whereas the early convection in the unsuccessful runs moved closer to the coast and had considerable portions located over the sea. This small displacement seems to be the main point deciding if the system decays or is able to live on and intensify into an MCS.

Although perhaps an extreme example, this case is in agreement with many previous studies pointing out the effects of small-scale variability in atmospheric parameters (e.g. Crook, 1996; Weckwerth, 2000). These results empha-

size the difficulty of forecasting the location and intensity of convective precipitation due to the chaotic nature of the atmosphere in convective weather events and the nonlinearity of the system with many feedbacks. In this case it is required to capture a chain of events that is dependent on precisely predicting the location of initial convection; only if the outflow of the initial convective system occurs in the right location can the damaging MCS be triggered.

The results of this work suggests that model domain shifting could be used to quantify how internal model variability contributes to the predictability of an event. However, this single case study needs to be expanded to cover more cases, for example in weather regimes with strong synoptic forcing and more stratiform precipitation and in other models such as ICON (ICOsahedral Non-hydrostatic) (Zängl et al., 2015). Moreover, whether changing the extent of domain shifting (e.g. from 1–10 grid points) is important should be evaluated.

*Data availability.* COSMO model output is available on request from the authors.

*Author contributions.* CB and AB both designed the numerical experiments, CB carried them out and performed the data analyses. Both authors contributed equally to the writing of the paper.

*Competing interests.* The authors declare that they have no conflict of interest.

*Acknowledgements.* The research leading to these results has been done within the subprojects B3 and B1 of the Transregional Collaborative Research Center SFB/TRR 165 "Waves to Weather" (www.wavestoweather.de) funded by the German Research Foundation (DFG). This work was performed on the computational resource ForHLR I funded by the Ministry of Science, Research and the Arts Baden-Württemberg and DFG ("Deutsche Forschungsgemeinschaft"). The authors wish to thank the Deutscher Wetterdienst (DWD) for providing the COSMO model code, the initial and boundary data as well as Radar-derived precipitation data. We are also grateful to U. Schättler (DWD) for his help in setting up the preprocessor int2lm of the COSMO model. Satellite pictures were kindly provided by Folke Olesen (IMK-ASF, KIT).

*Financial support.* This work was partly funded by the Deutsche Forschungsgemeinschaft (DFG, German Research Foundation) – Project-ID 257899354 – TRR 165.

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
