# Peer review of "Large impact of tiny model domain shifts for the Pentecost 2014 MCS over Germany"

_Weather and Climate Dynamics, 2019_

## Referee Comment (RC1) · Anonymous Referee #1 · 4 Oct 2019

This study examines simulations of a rather impactful MCS over Germany in 2014. The COSMO model at convection-allowing resolutions is used to simulate the MCS in a control run, and additional simulations move the model domain one grid point in eight cardinal directions. The predictability and forecast errors of the MCS are largely dependent on convection initiation in western France and subsequent propagation of the system over land, as well as the environment the system encounters during the day before impacting Germany. Substantial variability exists amongst the simulations, whereby some instances of the MCS-induced rainfall is forecast reasonably over Germany and other instances there is no precipitation in this region at all. It doesn't appear that systematic movement of the domain resulted in clustering of forecast errors either.

[Figure]

I do have some concerns regarding the motivation and methodology for this work. While the authors motivate the work as a worthwhile approach to improve convection-allowing model (CAM) forecasts, effectively creating sufficient spread across the forecast distribution, I struggle to see the benefit of this domain-shifting approach over other well-documented lateral boundary perturbation techniques (e.g., Torn et al. 2016), and I don't feel the author's sufficiently demonstrated this benefit in the introduction. Additionally, I believe the authors try to attribute the forecast variability to their methodology (i.e., domain shifting), but I do not feel there is substantial corroborating evidence to support this conclusion. For instance, the forecast variability and predictability could simply be a function of the dynamic sensitivity of the evolving forecast: MCSs developing along the coast of France are inherently less predictable. I have elaborated on these concerns below that should guide the authors in their manuscript revisions, outlined below in the "Major Comments" section. Other minor comments and technical suggestions follow the major comments.

Major Comments

1. My first and foremost major concern is the contribution of this work to the atmospheric science literature on MCSs, predictability, and forecast generation. I believe what the authors have described is merely a technique to perturb lateral boundary conditions, thereby producing spread in the initial states of the model simulations which filters into the forecasts of the MCS. If we assume the environment at the model boundaries is relatively homogeneous, at least within a few grid points which only amounts to < 10km, then what the authors have described here is essentially a 10-member ensemble forecast system with perturbed boundary conditions. This methodology is not in itself flawed by any means, but I do not believe it is innovative or new. See Torn et al. (2006), Gebhardt et al. (2011), and Romine et al. (2014) for other examples of boundary condition perturbation studies. Given this statement, however, I think the authors could easily address my concern by a number of avenues:

a. If the authors believe this truly is an innovative technique to generate CAM ensembles, they should either more succinctly clarify this in the introduction with references supporting this claim or demonstrate the methodology alongside some of the more traditional techniques (e.g., covariance perturbations) for this case study.

b. If the authors would still like to use the domain-shifting methodology to investigate the predictability of the MCS, I would caution attributing the methodology to why the MCS is inherently unpredictable. In order to scientifically attribute the poor forecast predictability to the domain-shifting methodology, substantially more analysis and simulations would need to be conducted. For instance, do you see the same poor predictability if the domain is moved 5, 10, or 20 grid points? What about if another perturbation technique is used? Can you reproduce the poor forecasts?

c. If the authors would rather focus on the predictability aspect of this event, I believe the authors could implement some other analysis techniques to derive some of the dynamic aspects for this case to complement what has been presented. Sensitivity approaches such as those demonstrated by Schumacher and Davis (2010) and Ancell and Hakim (2007) could be valuable additions to the analysis. I invite the authors to consult a number of papers that apply sensitivity analyses to convection-resolving forecasts as well: Bednarczyk and Ancell (2015), Hill et al. (2016), Limpert and Houston (2018), and Torn et al. (2016). Additionally, other aspects of predictability could be garnered through initializing ensemble forecasts at later times, which may answer the particular question of whether CI is the limiting factor of predictability.

2. A second concern I have is in the presentation of the forecast itself. There is no mention of the upper-level dynamics that could be supporting MCS development, particularly since the orientation of development and distribution of environmental parameters conducive to MCS propagation are misaligned from traditional understanding. For instance, the MCS propagation within a region of predominantly northwesterly or westerly surface winds, which would not advect the CAPE-rich air from the southeast. Typically, we would expect a convergence of moisture and higher theta-e air just ahead of the MCS, but this is not the case. Also, there is no mention as to what causes the

MCS to initiate so early in the day. My inclination from reading the forecast description is that the orientation of the longwave mid-tropospheric trough is supporting the traversal of short-wave troughs through western Europe. I suggest the authors add supporting evidence for how the MCS initiates, which could elucidate some other predictability elements that have not been considered, e.g. the position and placement of upper-level vorticity maxima.

3. Why is accumulated rainfall used as the sole metric of forecast evaluation? I would think observed radar reflectivity compared to simulated reflectivity would be a better metric for comparing model runs. Comparing reflectivity would better illuminate the intensity and structure of the MCS between observations and simulations; accumulated rainfall doesn't discriminate these differences well.

Minor Comments

Abstract

1. Why was the predictability low for the operational prediction systems? Do those systems parameterize convection or is it explicitly solved?

2. "However, the low predictability of the event was evident by the surprisingly large impact of tiny changes to the model domain": Is the argument there is low predictability because of dynamics or because of the model configuration? There appears to be two separate statements of predictability related to this event, but it is unclear what statements the authors really want to make. I'm assuming the main predictability element comes through the numerical (domain) aspect. Introduction

3. Lines 28-30: Were the German Weather Service operational models convection resolving? Is there any indication as to "why" the deterministic and ensemble systems failed to produce convection over Germany? This piece of discussion would be a good addition to the manuscript to help explain "why" the model forecasts failed and potentially motivate the use of convection-allowing models.

4. Line 35: First bullet point: what operational model is being discussed here? Second bullet point: Is the COSMO model being discussed here? Please be explicit about what model and associated configuration is being altered.

5. Line 49: What is COSMO-DE? While the COSMO acronym has been properly described, I don't know what "DE" references.

6. Lines 85-91: What benefit does "domain shifting" have over other traditional lateral boundary perturbation techniques (e.g., Torn et al. 2016)? I have not been convinced in the introduction that there is significant benefit in developing a new technique to perturb boundary conditions. Would it be appropriate to compare the described "domain shifting" technique with other perturbation techniques? Including this type of analysis would presumably shift the focus of your manuscript to an evaluation of ensemble-generation techniques for a specific MCS case study. Alternatively, the authors could instead focus on the true predictability of the event (rather than the domain shifting idea) and include some additional predictability analysis (e.g., ensemble sensitivity). See Schumacher and Davis (2010), Ancell and Hakim (2007), Bednarczyk and Ancell (2015), Hill et al. (2016), and Torn et al. (2016) for some examples of sensitivity analysis for precipitation and high-impact weather forecasts. (Major comment above)

7. I would suggest leaving the descriptive nouns out of the manuscript, and let the reader decide what is "surprising" or not (e.g., Line 92).

Section 4

1. What is the source of radar observations? Would be appropriate to add this into the manuscript for reproducibility.

2. Line 144: should be (Fig. 4a)

3. Lines 158-159: I actually do not agree with this statement. I think the reference forecast has some glaring errors that do not make this a particular good forecast. Consider revising or removing this statement.

4. Lines 179-181: All the eastward shift simulations have poorer prediction though.

5. Lines 270-272: I think a reasonable counter argument could be that the W run initiated the convection well to the east (east of the red circle) and therefore had an earlier impact over Germany than the reference run, making it a "poor" forecast. Additionally, this forward storm system appeared to greatly impact the CAPE field in Figure 9, which seemingly had an impact on the development of upstream convection in the red circle. Furthermore, there is clearly a neutral to slightly negatively-tilted mid-tropospheric trough to aid in the propagation of shortwaves (hard to tell where these might exist in the coarse resolution of Figure 1): what role did mid-tropospheric dynamics play in this system? I suggest a more thorough evaluation of the simulations and discussing all aspects of the environment more thoroughly, including any convection that might have influenced convection initiation (CI) in the focus area.

6. Line 284: The sea surface temperatures have not been described in detail yet. How do we know these SSTs are the limiting factor? We do not know what the SSTs from each simulation are or how they dynamically are impacting the simulation convection. Seems like a reaching statement without any evidence and I would suggest revising or providing more concrete, quantitative support.

References:

Ancell B. C. and G. J. Hakim, 2007: Comparing Adjoint- and Ensemble-Sensitivity Analysis with Applications to Observation Targeting. Monthly Weather Review, 135, 4117-4134

Bednarczyk C. N. and B. C. Ancell, 2015: Ensemble sensitivity analysis applied to a southern Plains convective event. Monthly Weather Review, 143, 230-249

Gebhardt C., S. E. Theis, M. Paulat, and Z. B. Bouallègue, 2011: Uncertainties in COSMO-DE precipitation forecasts introduced by model perturbations and variation of lateral boundaries. Atmospheric Research, 100, 168-177.

Hill A. J., C. C. Weiss, and B. C. Ancell, 2016: Ensemble senstivity analysis for mesoscale forecasts of dryline convection initiation. Monthly Weather Review, 144, 4161-4182.

Limpert G. L. and A. L. Houston, 2018: Ensemble senstivity analysis for targeted observations of supercell thunderstorms. Monthly Weather Review, 146, 1705-1721.

Romine G. S, C. S. Schwartz, J. Berner, K. R. Fossell, C. Snyder, J. L. Anderson, and M. L. Weisman, 2014: Representing forecast error in a convection-permitting ensemble system. Monthly Weather Review, 142, 4519-4541.

Schumacher R. S. and C. A. Davis, 2010: Ensemble-Based Forecast Uncertainty Analysis of Diverse Heavy Rainfall Events. Weather and Forecasting, 25, 1103-1122.

Torn. R., G. J. Hakim, and C. Snyder, 2006: Boundary conditions for limited-area ensemble Kalman filters. Monthly Weather Review, 134, 2490-2502.

Torn R., G. S. Romine, and T. J. Galarneau, Jr., 2016: Sensitivity of dryline convection forecasts to upstream forecast errors for two weakly forced MPEX cases. Monthly Weather Review, 145, 1831-1852.

---

## Referee Comment (RC2) · Anonymous Referee #2 · 17 Oct 2019

Weather Climate and Dynamics #: 2019-5 Title: Large impact of tiny model domain shifts for the Pentecost 2014 MCS over Germany Author(s): Christian Barthlott and Andrew I. Barrett Review completed: 10/17/2019

This study presents the results of a simple experiment to test the sensitivity in simulating a high impact precipitation event over Germany by shifting the model domain by seemingly inconsequential amounts. While the study focuses on the impact of the domain shifts, essentially the ensemble model setup is an exercise in perturbing the initial conditions/lateral boundaries and thus the intrinsic predictability of convective storms.

The main result of the ensemble was that members that initialized convection in France

and then subsequently moved over cooler, ocean air resulted in weaker convection that dissipated before being able to intensify into the observed MCS in Germany. On the other hand, members that kept the convection over land where it was able to tap into a more favorable environment produced convective systems that were reasonably well forecasted over Germany.

In general, this paper needs provide a clearer link to previous studies that have investigated the impact of perturbing the initial conditions/lateral boundaries. I am still unconvinced that shifting the domain would be a more promising avenue to "account for uncertainties in the initial and boundary conditions" than other techniques (see the work from Ryan Torn and colleagues since the mid-2000s). Additionally, I believe more analysis is needed than a cursory comparison of precipitation and environmental parameters. What preempted the deviations in convection evolution over land/sea? Plots and discussions of differences in upper-level vorticity, MSLP, and even SSTs would improve the analysis. Once these two chief concerns have been addressed, I will provide a more thorough review, including specific comments and suggestions, prior to publication.

---

## Author Comment (AC2) · 26 Nov 2019

**Responses to the reviewers**

Large impact of tiny model domain shifts for the Pentecost 2014 MCS over Germany

by Christian Barthlott and Andrew I. Barrett                    November 26, 2019
* * *
Dear Editor,

This letter accompanies our revised manuscript. We are grateful for the reviewer's helpful comments, and hope our revision addresses them all. Below we detail the changes made in our revision. We include the text of the reviews in black, our responses are in blue.

**Reviewer 2**

This study presents the results of a simple experiment to test the sensitivity in simulating a high impact precipitation event over Germany by shifting the model domain by seemingly inconsequential amounts. While the study focuses on the impact of the domain shifts, essentially the ensemble model setup is an exercise in perturbing the initial conditions/lateral boundaries and thus the intrinsic predictability of convective storms. The main result of the ensemble was that members that initialized convection in France and then subsequently moved over cooler, ocean air resulted in weaker convection that dissipated before being able to intensify into the observed MCS in Germany. On the other hand, members that kept the convection over land where it was able to tap into a more favorable environment produced convective systems that were reasonably well forecasted over Germany.

In general, this paper needs provide a clearer link to previous studies that have investigated the impact of perturbing the initial conditions/lateral boundaries. I am still unconvinced that shifting the domain would be a more promising avenue to "account for uncertainties in the initial and boundary conditions" than other techniques (see the work from Ryan Torn and colleagues since the mid-2000s). Additionally, I believe more analysis is needed than a cursory comparison of precipitation and environmental parameters. What preempted the deviations in convection evolution over land/sea? Plots and discussions of differences in upper-level vorticity, MSLP, and even SSTs would improve the analysis. Once these two chief concerns have been addressed, I will provide a more thorough review, including specific comments and suggestions, prior to publication.

- The first point we want to address is similar to the reply to the first and second comment of Reviewer #1. We do not believe that our technique is a new method to generate ensembles with perturbed initial/boundary conditions in operational convective-scale ensemble forecasting. However, we were surprised to see such a large influence of these tiny changes on the simulation results and strongly believe that this method should be tested for more cases (also with different extents of domain shifting) and other models. It may also be that the high sensitivity is a feature of days with low predictability only, which would be a useful information to have. Therefore, a more systematic evaluation is left for future work. The goal of this paper was not to assess the impact of other perturbation techniques, as we already mentioned the poor forecast quality of the operational COSMO-DE-EPS of the German Weather Service in the introduction. It is of special interest to see, if other cases with low predictability (i.e. forecast busts) show the same sensitivity. However, we think that such an analysis would not fit into the present paper and is therefore left for future work. For these reasons, we did not refer much to other techniques to perturb initial and boundary conditions in our manuscript.

  **Changes to paper**

Abstract:

This study demonstrates the potentially huge impact of tiny model domain shifts on forecasting convective processes in this case, which suggests that  *the sensitivity to similarly small initial condition perturbations* should be evaluated  *across other cases, model* and weather regimes.

Summary:

The results of this work suggests that  model domain shifting could be used to  *quantify how* uncertainties in the initial and boundary conditions  *contribute to the predictability of an event.* However, this single case study needs to be expanded to cover more cases  ,for example in weather regimes with strong synoptic forcing and more stratiform precipitation and in other models such as ICON...

- The upper-level dynamics are similar in all model runs in the early stage of the convection over France. Hoskins et al. (1978) demonstrated that the traditional form of the quasi-geostrophic omega equation can be rewritten using the Q-vector and that regions of upward (downward) vertical motion are associated with Q-vector convergence (divergence). In Fig. R.1, there are no noticable differences in the Q-vector divergence, nor does the model simulate any variations in geopotential height. This indicates that the large scale forcing is similar for these model runs. We included a statement on that at the end of section 4.6.

[Figure]

Figure R.1: Q-vector divergence (colours), 500 hPa geopotential height (contour lines), and precipitation rate (hatched) for the W run (left) and the REF run (right) at 0800 UTC.

**Changes to paper**

*"In addition to this analysis, we further want to point out that the upper-level dynamics are similar in all model runs in the early stage of the convection over France. Hoskins (1978) showed that the traditional form of the quasi-geostrophic omega equation can be rewritten using the Q-vector and that regions of upward (downward) vertical motion are associated with Q-vector convergence (divergence). We calculated the divergence of the Q-vector at 500 hPa and found no noticable differences between successful and unsuccessful runs, nor does the model simulate any variations in geopotential height (not shown). This indicates that the large scale forcing is similar for these model runs and not responsible for the simulation result differences."*

- We also analysed SST as suggested. Our reply is the same as for Reviewer #1 (minor comment 6): The surface temperature and CAPE are depicted in Fig. R.2. The sea surface temperature

is much lower than the land surface temperature, at least in the northwestern coast of France where no significant amounts of rain was simulated in the last hours. As a result of these lower temperatures, CAPE is significantly reduced over sea. Along the coastline, there is a strong gradient in temperature (23 → 15 deg C) and CAPE. These statements also hold true for the remaining model runs. We added some remarks on that in the text, but decided not to provide an extra figure.

[Figure]

Figure R.2: Surface temperature of the REF run at 1000 UTC (colours, in deg C) and CAPE (white contours, in $\mathrm{J\,kg^{-1}}$).

**Changes to paper:**
*"The sea surface temperatures along the French coast lie around 15℃ and are much lower than the land surface temperatures (around 23℃, not shown). This temperature distribution is similar in all model runs for the preconvective environment."*

- Furthermore, we conducted an ensemble sensitivity analysis and included those results in a new section.

**Changes to paper:**
We include the sensitivity analysis in a new section 4.5, this includes explanation of the ensemble sensitivity analysis method and interpretation of the results and includes discussion of the newly added Figure 9.

**Additional changes to the paper:**

1. We included a new sentence in the introduction about two recent papers:

   *"Recent studies of Schneider et al. (2019) and Keil et al. (2019) have also shown that different assumptions for the amount of cloud condensation nuclei could be included in convective-scale ensemble forecasting, but only if the model employs a double-moment microphysics scheme."*

2. Old Figure 9 was enhanced by increasing the size, length, and density of the wind arrows.

3. Information about the financial support was added.

---

## Author Comment (AC1)

**Responses to the reviewers**

Large impact of tiny model domain shifts for the Pentecost 2014 MCS over Germany

by Christian Barthlott and Andrew I. Barrett                    November 26, 2019
* * *
Dear Editor,

This letter accompanies our revised manuscript. We are grateful for the reviewer's helpful comments, and hope our revision addresses them all. Below we detail the changes made in our revision. We include the text of the reviews in black, our responses are in blue.

**Reviewer 1**

This study examines simulations of a rather impactful MCS over Germany in 2014. The COSMO model at convection-allowing resolutions is used to simulate the MCS in a control run, and additional simulations move the model domain one grid point in eight cardinal directions. The predictability and forecast errors of the MCS are largely dependent on convection initiation in western France and subsequent propagation of the system over land, as well as the environment the system encounters during the day before impacting Germany. Substantial variability exists amongst the simulations, whereby some instances of the MCS-induced rainfall is forecast reasonably over Germany and other instances there is no precipitation in this region at all. It doesn't appear that systematic movement of the domain resulted in clustering of forecast errors either.

Major Comments

1. My first and foremost major concern is the contribution of this work to the atmospheric science literature on MCSs, predictability, and forecast generation. I believe what the authors have described is merely a technique to perturb lateral boundary conditions, thereby producing spread in the initial states of the model simulations which filters into the forecasts of the MCS. If we assume the environment at the model boundaries is relatively homogeneous, at least within a few grid points which only amounts to < 10km, then what the authors have described here is essentially a 10-member ensemble forecast system with perturbed boundary conditions. This methodology is not in itself flawed by any means, but I do not believe it is innovative or new. See Torn et al. (2006), Gebhardt et al. (2011), and Romine et al. (2014) for other examples of boundary condition perturbation studies. Given this statement, however, I think the authors could easily address my concern by a number of avenues:

    a. If the authors believe this truly is an innovative technique to generate CAM ensembles, they should either more succinctly clarify this in the introduction with references supporting this claim or demonstrate the methodology alongside some of the more traditional techniques (e.g., covariance perturbations) for this case study.

    We do not believe that our technique is a new method to generate ensembles with perturbed initial/boundary conditions in operational convective-scale ensemble forecasting. However, we were surprised to see such a large influence of these tiny changes on the simulation results and strongly believe that this method should be tested for more cases (also with different extents of domain shifting) and other models. It may also be that the high sensitivity is a feature of days with low predictability only, which would be a useful information to have. Therefore, a more systematic evaluation is left for future work. We adapted the text to make that clearer.

**Changes to paper**

Abstract:

This study demonstrates the potentially huge impact of tiny model domain shifts on forecasting convective processes in this case, which suggests that  *the sensitivity to similarly small initial condition perturbations* should be evaluated  *across other cases, model* and weather regimes.

Summary:

The results of this work suggests that  model domain shifting could be used to  *quantify how* uncertainties in the initial and boundary conditions  *contribute to the predictability of an event*. However, this single case study needs to be expanded to cover more cases  ,for example in weather regimes with strong synoptic forcing and more stratiform precipitation and in other models such as ICON...

b. If the authors would still like to use the domain-shifting methodology to investigate the predictability of the MCS, I would caution attributing the methodology to why the MCS is inherently unpredictable. In order to scientifically attribute the poor forecast predictability to the domain-shifting methodology, substantially more analysis and simulations would need to be conducted. For instance, do you see the same poor predictability if the domain is moved 5, 10, or 20 grid points? What about if another perturbation technique is used? Can you reproduce the poor forecasts?

*The goal of this paper was not to assess the impact of other perturbation techniques, as we already mentioned the poor forecast quality of the operational COSMO-DE-EPS of the German Weather Service in the introduction. While trying to find a model setup which could reproduce the MCS, we made a lot of tests, also with respect to domain size and domain location. When we changed the location of the domain by 2, 10, and 20 grid points, we already had successful and unsuccessful results. This is why we went to the minimal domain shifting possible, namely 1 grid point in eight cardinal directions. We believe that this is a good first step and this method should be evaluated as mentioned in the reply to the first comment. It is of special interest to see, if other cases with low predictability (i.e. forecast busts) show the same sensitivity. However, we think that such an analysis would not fit into the present paper and is therefore left for future work.*

**Changes to paper:**
*none*

c. If the authors would rather focus on the predictability aspect of this event, I believe the authors could implement some other analysis techniques to derive some of the dynamic aspects for this case to complement what has been presented. Sensitivity approaches such as those demonstrated by Schumacher and Davis (2010) and Ancell and Hakim (2007) could be valuable additions to the analysis. I invite the authors to consult a number of papers that apply sensitivity analyses to convection-resolving forecasts as well: Bednarczyk and Ancell (2015), Hill et al. (2016), Limpert and Houston (2018), and Torn et al. (2016). Additionally, other aspects of predictability could be garnered through initializing ensemble forecasts at later times, which may answer the particular question of whether CI is the limiting factor of predictability.

*We thank the reviewer for this useful hint and performed an ensemble sensitivity analysis*

for our model runs. We present the results in the new section 4.5.

**Changes to paper:**

We include the sensitivity analysis in a new section 4.5, this includes explanation of the method and interpretation of the results and includes discussion of the newly added Figure 9.

2. A second concern I have is in the presentation of the forecast itself. There is no mention of the upper-level dynamics that could be supporting MCS development, particularly since the orientation of development and distribution of environmental parameters conducive to MCS propagation are misaligned from traditional understanding. For instance, the MCS propagation within a region of predominantly northwesterly or westerly surface winds, which would not advect the CAPE-rich air from the southeast. Typically, we would expect a convergence of moisture and higher theta-e air just ahead of the MCS, but this is not the case. Also, there is no mention as to what causes the MCS to initiate so early in the day. My inclination from reading the forecast description is that the orientation of the longwave mid-tropospheric trough is supporting the traversal of short-wave troughs through western Europe. I suggest the authors add supporting evidence for how the MCS initiates, which could elucidate some other predictability elements that have not been considered, e.g. the position and placement of upper-level vorticity maxima.

The MCS propagation has not been the subject of the paper so far, as we focused more on the fact why the precursors of the MCS dissipate or not. This is why Figure 9 only presents the period between 1000 and 1400 UTC. The system later evolves into an MCS as can be seen in Fig. R.1, which shows that the distribution of environmental parameters are not misaligned from traditional understanding. We observe exactly what the reviewer has anticipated, but was

[Figure]

Figure R.1: Convective available potential energy (colour shading, in $\mathrm{J\,kg^{-1}}$, 30-min precipitation rate (blue colour shading, in $\mathrm{mm\,(30\,min)^{-1}}$, and 10-m wind field (arrows) between 1600–2100 UTC on 9 June. Gray areas indicate low-level wind convergence larger than $0.35\cdot10^{-3}$ $\mathrm{m\,s^{-1}}$ and hatched areas represent regions where convective inhibition is smaller than $5\,\mathrm{J\,kg^{-1}}$.

not able to see in the Figure 9: an advection of CAPE-rich air from the East, with well-defined region of low-level wind convergence at the outflow boundary. The MCS cleary moves into the region with high CAPE which corresponds to high values of equivalent potential temperature.

We included this Figure in a new subsection 4.7 in the manuscript.

**Changes to paper:**
new Figure 11 and new subsection 4.7:

*"Having established a possible explanation for the decay of the precursors of the MCS in the previous section, we now analyze the further evolution of the system into a MCS using the reference simulation (Fig. 11). To the east of the system, the model simulates an east-west oriented region of high low-level equivalent potential temperature in the north-central part of Germany, which corresponds to CAPE values between 3000–4000 $J\,kg^{-1}$. This CAPE-rich air is advected with easterly winds towards the convective system over the Netherlands. Colliding with the cell's outflow, a strong low-level mass and moisture convergence occurs, which fosters the evolution into a MCS. As already discussed in section 4.4, the 0-6 km deep layer shear shows suitable conditions for highly-organised convection (27–30 $m\,s^{-1}$). The maximum rain intensities reach locally up to $22\,mm\,(30\,min)^{-1}$ with a weakly defined bow-like structure of precipitation, typical of storms with an intense rear-inflow jet. In the wake of the MCS, CAPE is almost entirely consumed. From 23:00 UTC onwards, the MCS is decaying while further travelling towards Poland (not shown)."*

3. Why is accumulated rainfall used as the sole metric of forecast evaluation? I would think observed radar reflectivity compared to simulated reflectivity would be a better metric for comparing model runs. Comparing reflectivity would better illuminate the intensity and structure of the MCS between observations and simulations; accumulated rainfall doesn't discriminate these differences well.

We believe that rainfall at the ground is a suitable metric to assess the sensitivity of the model in simulating an MCS. Even if the model shows some discrepancies with respect to location and propagation speed, an overall good agreement between simulation and observation exists. Moreover, the focus of our paper lies on the sensitivity of the model to domain shifting and in-depth comparison of the MCS of the reference run with radar observations is not necessary for the reader to follow our story. Also, radar reflectivities are not available to the authors and the simulations would have to be done again with a radar-forward operator. Having said that, we think that an evaluation with rainfall is sufficient for our purpose.

Minor Comments:

1. Why was the predictability low for the operational prediction systems? Do those systems parameterize convection or is it explicitly solved?

The origin of the low predictability of this case was unknown so far. This paper is another contribution to that topic. While in the Barthlott et al. (2017) paper, an enlargement of the model domain, a higher grid spacing and a single/double moment microphysics scheme were addressed, this study has shown show that small displacements of the convective system over France can lead to a decaying system or to a system developing into an MCS later on. The operational prediction systems used the same grid spacing as in our study, so deep convection was resolved and shallow convection parameterized. We added remarks in the model description and in the summary.

**Changes to paper**
Model description:

Deep convection is resolved explicitly and a modified Tiedtke-scheme (Tiedtke, 1989) is used to parameterize shallow convection *(as did the operational deterministic and ensemble prediction system at that time)*.

Summary:

However, the predictability of this event was very low; neither the operational deterministic nor the ensemble prediction system *(both convection resolving)* captured the event with more than 12 hours lead time.

2. "However, the low predictability of the event was evident by the surprisingly large impact of tiny changes to the model domain": Is the argument there is low predictability because of dynamics or because of the model configuration? There appears to be two separate statements of predictability related to this event, but it is unclear what statements the authors really want to make. I'm assuming the main predictability element comes through the numerical (domain) aspect.

   As outlined in the reply to the previous comment, we do not know the origin of the low predictability. In our study, the low predictability is reflected by the domain effect.

   **Changes to paper:**
   none

3. Introduction Lines 28-30: Were the German Weather Service operational models convection resolving? Is there any indication as to "why" the deterministic and ensemble systems failed to produce convection over Germany? This piece of discussion would be a good addition to the manuscript to help explain "why" the model forecasts failed and potentially motivate the use of convection-allowing models.

   The operational models were convection-resolving, i.e. with a horizontal grid spacing of 2.8 km. As already mentioned earlier, we do not know why these model runs failed to produce convection over Germany. We added two remarks in the manuscript about this fact and the operational resolution.

   **Changes to paper**
   Model description:
   Deep convection is resolved explicitly and a modified Tiedtke-scheme (Tiedtke, 1989) is used to parameterize shallow convection *(as did the operational deterministic and ensemble prediction system at that time)*.

   Summary:
   However, the predictability of this event was very low; neither the operational deterministic nor the ensemble prediction system *(both convection resolving)* captured the event with more than 12 hours lead time.

4. Line 35: First bullet point: what operational model is being discussed here? Second bullet point: Is the COSMO model being discussed here? Please be explicit about what model and associated configuration is being altered.

   We did not mean the operational model, but the COSMO model in an operational setup. We modified the text to make that clearer.

   **Changes to paper:**
   A series of different numerical simulations for the convective events of 8 and 9 June 2014 were performed *"with the COSMO model"*, the main findings were:

   - The  *COSMO model (in quasi operational set-up, without data assimilation)* initialized at 00:00 UTC reproduced the events on 8 June only, but not the mesoscale convective system (MCS) on 9 June.

5. Line 49: What is COSMO-DE? While the COSMO acronym has been properly described, I don't know what "DE" references.

COSMO-DE is the name of operational configuration at DWD over Germany. This information is not needed here. So instead of explaining it, we just replaced *"COSMO-DE domain"* with *"model domain"*.

6. Lines 85-91: What benefit does "domain shifting" have over other traditional lateral boundary perturbation techniques (e.g., Torn et al. 2016)? I have not been convinced in the introduction that there is significant benefit in developing a new technique to perturb boundary conditions. Would it be appropriate to compare the described "domain shifting" technique with other perturbation techniques? Including this type of analysis would presumably shift the focus of your manuscript to an evaluation of ensemble-generation techniques for a specific MCS case study. Alternatively, the authors could instead focus on the true predictability of the event (rather than the domain shifting idea) and include some additional predictability analysis (e.g., ensemble sensitivity). See Schumacher and Davis (2010), Ancell and Hakim (2007), Bednarczyk and Ancell (2015), Hill et al. (2016), and Torn et al. (2016) for some examples of sensitivity analysis for precipitation and high-impact weather forecasts. (Major comment above)

It was not our goal to evaluate different ensemble-generation techniques. Our case study is a first step, but needs evaluation with more cases before it can be compared to different methods to introduce uncertainties in the initial and boundary conditions. However, the study of Henneberg et al. (2018) showed, that by shifting the model domain, by ten to 30 grid points, an estimate of the uncertainty of the model results can be achieved with a sufficient large model spread. We believe that the large impact of these tiny changes need further evaluation with more cases, different extents of domain shifting, and other models. In several places in the manuscript, we now state that this method needs further evaluation and that suitability for representing uncertainties should be compared to traditional lateral boundary perturbation techniques.

We are grateful for the examples of the sensitivity analysis. We performed such an analysis, the results are presented in the new section 4.5.

**Changes to paper:**
We include the sensitivity analysis in a new section 4.5, this includes explanation of the method and interpretation of the results and includes discussion of the newly added Figure 9.

7. I would suggest leaving the descriptive nouns out of the manuscript, and let the reader decide what is "surprising" or not (e.g., Line 92).

We do believe that our technique of domain shifting of just 1 grid point provides surprising or at least unexpected results for this particular case. Given the large model domain and the minor changes at the boundaries, we would not have anticipated such a large dependency. Therefore we like to keep our phrasing in the current form.

**Changes to paper:**
none

Section 4

1. What is the source of radar observations? Would be appropriate to add this into the manuscript for reproducibility.

The radar observation come from the radar network of the German Weather Service, the product is called RADOLAN. We added this sentence at the beginning of section 4.1.

**Changes to paper**

*"Here we compare our simulations to radar-derived precipitation from the precipitation analysis algorithm RADOLAN (Radar Online Adjustment), which combines weather radar data with hourly surface precipitation observations of about 1300 automated rain gauges to get quality-controlled, high-resolution (1 km) quantitative precipitation estimations."*

2. Line 144: should be (Fig. 4a)

   Done

3. Lines 158-159: I actually do not agree with this statement. I think the reference forecast has some glaring errors that do not make this a particular good forecast. Consider revising or removing this statement.

   We agree with the reviewer that the bow-like structure is not well-defined in our simulations. We therefore removed that sentence. Otherwise, we think that the model is doing a reasonable job, despite the differences already described in the text.

   **Changes to paper:**

4. Lines 179-181: All the eastward shift simulations have poorer prediction though.

   We agree with the reviewer as the precipitation in the E-run is more to the North and does not extend as much to the East as the other successful runs.

   **Changes to paper:**
   *"However, as the precipitation in the E run is more to the North and does not extend as much to the East as the other successful runs, all the eastward shift simulations have poorer prediction."*

5. Lines 270-272: I think a reasonable counter argument could be that the W run initiated the convection well to the east (east of the red circle) and therefore had an earlier impact over Germany than the reference run, making it a "poor" forecast. Additionally, this forward storm system appeared to greatly impact the CAPE field in Figure 9, which seemingly had an impact on the development of upstream convection in the red circle. Furthermore, there is clearly a neutral to slightly negatively-tilted mid-tropospheric trough to aid in the propagation of short-waves (hard to tell where these might exist in the coarse resolution of Figure 1): what role did mid-tropospheric dynamics play in this system? I suggest a more thorough evaluation of the simulations and discussing all aspects of the environment more thoroughly, including any convection that might have influenced convection initiation (CI) in the focus area.

   The reviewer is right about the fact that in the W run, a convective cell occurs east of the red circle. We already mentioned this at the end of section 4.5:

   *"The isolated cell, to the north west of these plots between 1000-1100 UTC, does not appear to be important to the decay of the cell of interest. It is located approximately 150 km upstream. The cell is stronger in the W run leading to a slight reduction of CAPE and therefore creating slightly less favorable environmental conditions in the area into which the main cell would later move. However, it appears that the weakening of the main cell occurred independently of the cell upstream and can rather be attributed to the proximity to the colder sea surface."*

   Our main counter argument would be that this convection does indeed reduce the CAPE locally, but the reduction in CAPE does not reach the region where the cell of interest is decaying. At 1130 UTC, the cell in the W run is decaying although further downstream there is still a tongue

of air with higher CAPE values as in the REF run. Moreover, in the SE run there is no cell to the east of the system of interest, and convection dies out anyway, in spite of the unaltered CAPE field downstream. We therefore conclude that the weakening of the main cell occurred independently of the cell upstream and can rather be attributed to the proximity to the colder sea surface.

**Changes to paper:**

none

6. Line 284: The sea surface temperatures have not been described in detail yet. How do we know these SSTs are the limiting factor? We do not know what the SSTs from each simulation are or how they dynamically are impacting the simulation convection. Seems like a reaching statement without any evidence and I would suggest revising or providing more concrete, quantitative support.

The surface temperature and CAPE are depicted in Fig. R.2. The sea surface temperature is much lower than the land surface temperature, at least in the northwestern coast of France where no significant amounts of rain was simulated in the last hours. As a result of these lower temperatures, CAPE is significantly reduced over sea. Along the coastline, there is a strong gradient in temperature (23 → 15 deg C) and CAPE. These statements also hold true for the remaining model runs. We added some remarks on that in the text, but decided not to provide an extra figure.

[Figure]

Figure R.2: Surface temperature of the REF run at 1000 UTC (colours, in deg C) and CAPE (white contours, in $J\,kg^{-1}$).

**Changes to paper:**

*"The sea surface temperatures along the French coast lie around 15℃ and are much lower than the land surface temperatures (around 23℃, not shown). This temperature distribution is similar in all model runs for the preconvective environment."*

References:

Ancell B. C. and G. J. Hakim, 2007: Comparing Adjoint- and Ensemble-Sensitivity Analysis with

Applications to Observation Targeting. Monthly Weather Review, 135, 4117-4134

Bednarczyk C. N. and B. C. Ancell, 2015: Ensemble sensitivity analysis applied to a southern Plains convective event. Monthly Weather Review, 143, 230-249

Gebhardt C., S. E. Theis, M. Paulat, and Z. B. Bouallgue, 2011: Uncertainties in COSMO-DE precipitation forecasts introduced by model perturbations and variation of lateral boundaries. Atmospheric Research, 100, 168-177.

Hill A. J., C. C. Weiss, and B. C. Ancell, 2016: Ensemble senstivity analysis for mesoscale forecasts of dryline convection initiation. Monthly Weather Review, 144, 4161-4182.

Limpert G. L. and A. L. Houston, 2018: Ensemble senstivity analysis for targeted observations of supercell thunderstorms. Monthly Weather Review, 146, 1705-1721.

Romine G. S, C. S. Schwartz, J. Berner, K. R. Fossell, C. Snyder, J. L. Anderson, and M. L. Weisman, 2014: Representing forecast error in a convection-permitting ensemble system. Monthly Weather Review, 142, 4519-4541.

Schumacher R. S. and C. A. Davis, 2010: Ensemble-Based Forecast Uncertainty Analysis of Diverse Heavy Rainfall Events. Weather and Forecasting, 25, 1103-1122.

Torn. R., G. J. Hakim, and C. Snyder, 2006: Boundary conditions for limited-area ensemble Kalman filters. Monthly Weather Review, 134, 2490-2502.

Torn R., G. S. Romine, and T. J. Galarneau, Jr., 2016: Sensitivity of dryline convection forecasts to upstream forecast errors for two weakly forced MPEX cases. Monthly Weather Review, 145, 1831-1852.

**Additional changes to the paper:**

1. We included a new sentence in the introduction about two recent papers:

   *"Recent studies of Schneider et al. (2019) and Keil et al. (2019) have also shown that different assumptions for the amount of cloud condensation nuclei could be included in convective-scale ensemble forecasting, but only if the model employs a double-moment microphysics scheme."*

2. Old Figure 9 was enhanced by increasing the size, length, and density of the wind arrows.

3. Information about the financial support was added.

---

## Author Response (AR2)

**Responses to the reviewers**

Large impact of tiny model domain shifts for the Pentecost 2014 MCS over Germany

by Christian Barthlott and Andrew I. Barrett                    April 2, 2020

Dear Editor,

This letter accompanies our revised manuscript. We are grateful for the helpful comments, and hope our revision addresses them all. Below we detail the changes made in our revision. We include the text of the review in black, our responses are in blue.

**Co-Editor comments**

The reviewer's main criticism refers to the motivation of your work and its connection to the relevant literature. I agree with the reviewer that "reporting on the surprisingly large sensitivity" of your domain shifting technique is a rather weak motivation for a scientific study, and I'd ask to formulate a specific research question or hypothesis that you're addressing in your study. For instance, if this hypothesis should be that small shifts in the domain can have huge impacts on the prediction skill, than it would be natural to discuss previous studies on the impacts of changed boundary conditions (using other techniques) and also on small (stochastic?) perturbations in the initial conditions in the introduction. Possibly this could be fit into the manuscript by shortening the detailed description of your own previous research on the MCS event. Furthermore, also the discussion in the last section of the paper could be extended along this line. For example, I'm wondering how you can conclude that "model domain shifting could be used to quantify how uncertainties in the initial and boundary conditions contribute to the predictability" without comparing the sensitivity with other techniques (the spread generated by your technique may still be smaller and thus underestimate the effect of these uncertainties, which wouldn't be a proper quantification in this case).

It was not our original intention to introduce a new method to generate ensembles with perturbed initial/boundary conditions in operational convective-scale ensemble forecasting. Our main goals were (i) to produce a reasonable forecast of this major event by having a model domain large enough to cover all stages of the event and (ii) investigate the impact of the domain size and location (i.e. IBC/LBC) on the simulation results. Due to the large influence of these tiny changes on the simulation results, we strongly believe that this method should be tested for more cases (also with different extents of domain shifting) and other models. It may also be that the high sensitivity is a feature of days with low predictability only, which would be a useful information to have. Therefore, a more systematic evaluation is left for future work. We rephrased our introduction to better motivate this study.

We also followed the suggestion of the reviewer and conducted an additional model run on the reference domain with perturbed lateral boundary conditions. The result of this run together with the requested comments on other boundary-perturbation techniques is now included in a separate section 5 at the end of the paper.

**Reviewer 1**

**Major Comments**

**1. Motivation**

The technique used by the authors is new to the literature on convection-allowing ensemble forecasts. However, there is no clear motivation as to why this technique is preferred over other boundary-perturbation techniques (e.g., Torn et al. 2006). Both I and Reviewer #2 saw this as a flaw in the previous manuscript and it has not been addressed within the introduction. I do not expect the authors to complete companion analyses using other techniques to generate ensemble forecasts (although I

would like to see those comparisons), but I do expect the authors to at the very least discuss the advantages and disadvantages of the various techniques (including their own) which motivates this work. There is no reason to reinvent the wheel in ensemble generation unless sufficient reasoning motivates a more simple technique. What benefit does this technique have over others that make it worthwhile to pursue?

The technique of domain shifting is not new, we already mentioned the study of Henneberg et al. (2018) in the introduction. However, in their study, the simulation domain was considerably smaller and the perturbations were introduced by shifting the domain boundaries by ten to 30 grid points north and eastwards. Our study is the first one to address all 8 directions by just one grid point on a large model domain.

It was not our original intention to introduce a new method to generate ensembles with perturbed initial/boundary conditions in operational convective-scale ensemble forecasting. Our main goals were (i) to produce a reasonable forecast of this major event by having a model domain large enough to cover all stages of the event and (ii) investigate the impact of the domain location (i.e. IBC/LBC) on the simulation results. Due to the large influence of these tiny changes on the simulation results, we strongly believe that this method should be tested for more cases (also with different extents of domain shifting) and other models. It may also be that the high sensitivity is a feature of days with low predictability only, which would be a useful information to have. Therefore, a more systematic evaluation is left for future work.

We also followed the suggestion further below and conducted an additional model run on the reference domain with perturbed lateral boundary conditions. The result of this run together with the requested comments on other boundary-perturbation techniques is now included in a separate section 5 at the end of the paper.

**Changes to paper:**
We rephrased our introduction at several places to better motivate this study and included a new section 5 towards the end of the paper. Parts of the introduction were also moved to this new section, please see the track-changed version for all changes.

One possible new avenue to pursue is comparing the conducted experiments with parallel forecasts that use small, stochastic perturbations on the boundary conditions of the reference forecast. I suspect that the forecast variability seen in the current forecasts is simply the result of numerical noise and chaos seeding (see Ancell et al. 2018), whereby small perturbations in remote locations of the domain (e.g., the boundary) can induce very large changes in forecast precipitation for unrealistic reasons; the perturbations propagate during each model timestep the distance of the finite differencing scheme, which is typically faster than the speed of sound. The authors could actually use this companion analysis to prove or disprove that their methodology is sound and actually producing meaningful spread in their forecasts, i.e. represents true predictability.

We thank the reviewer for this suggestion and performed another model run on the reference domain, but with stochastic perturbations on the boundary conditions. We added random temperature fluctuations with a Gaussian distribution of zero mean and a standard deviation of 0.01 K at all levels for the model boundaries. The results of this run together with comments on other perturbation techniques is now in a new section 5. In this paper, we wanted to demonstrate the large sensitivity of the model results to a relatively simple technique of perturbing the initial and boundary conditions. The applicability of this technique has to be tested for other (and more) cases and could then also be compared to other operational used methods. Thus, we mentioned a number of other methods in the new section, but did not provide a detailed intercomparison of different methods or rating of our method against others. We believe that this would blow up the paper unnecessarily and should be

done in the mentioned follow-up work.

**Changes to paper:**
New section 5 and modifications in the Summary/Conclusion section

**2. Ensemble Sensitivity**

My previous suggestion to use ensemble sensitivity was motivated by wanting to see that the reasoning for forecast differences proposed by the authors (e.g., closer proximity to cold SSTs) could be deduced statistically, and not just subjectively by the authors. For instance, does higher precipitation in Germany during the 1700-0000 UTC timeframe (Fig. 4) statistically relate to higher/lower CAPE, stronger convergence, or upper-level dynamics at an earlier time that can explain the differences in the simulations? The analysis presented is a good start, but maybe insufficient to answer the dynamic questions about what is influencing the forecasts. I suggest the authors consider applying ensemble sensitivity to the fields in Figure 8 to support their claims about what ultimately influences MCS initiation and propagation into Germany.

We thank the reviewer for clarifying the motivation behind the suggestion of ensemble sensitivity analysis. Following their suggestion, Figure 9 has been replaced with a new figure looking at some of the variables from Figure 8 and now with a consistent time range to Figure 10. As is now shown, CAPE and precipitation rate do correlate well with later success of the simulation. However, the ensemble sensitivity analysis cannot be used to describe meaningful differences in the simulations before the convection has formed and organised. Only once it is organised is the sensitivity to CAPE and precipitation rate clear. The initial causes of the different tracks of the convective cells remain unclear but are likely chaotic in origin.

Most variables in Figure 8 only show a difference between successful and unsuccessful simulations at and after 12 UTC. CAPE and precipitation rate are examples of these - and the sensitivity is shown clearly in Figure 9. Low-level convergence and the resulting vertical velocity at 500 m both show differences before 12 UTC; however, Figure 9 shows that the location of these differences are not systematic in the set of simulations as the ensemble sensitivity analysis is largely dominated by noise until after 12 UTC.

Ensemble sensitivity analysis fields of CAPE and CIN were very similar and CIN is therefore omitted. Similarly, precipitable water is very similar to precipitation rate in highlighting the location of the existing convective cell. There really are no large-scale differences in e.g. CAPE, which can also be seen in Figure 10. We believe that the current ensemble sensitivity analysis has proved valuable in highlighting systematic differences between the simulations, but that further analysis will not allow the origin of the differences to be uncovered - largely because of the disparate locations of the initial convection.

**Changes to paper:**
Figure 9 has been replaced and improved by using different variables consistent with Figure 8. The text in section 4.4 has been modified consistently with the new figure.

**3. Forecast Differences**

It may be helpful for the eventual readership to actually see forecast difference plots that illustrate the reduced CAPE near the french coastline, enhanced convergence, etc. rather than try to interpret the differences based on equivalent environmental plots. For instance, in Figure 10 the differences between the W, REF, and SE precipitation rates are fairly obvious, but differences in the CAPE fields (and presumably a number of other environmental parameters) are much more difficult to make out. Rather than showing identical forecast plots side-by-side as in Figs. 10, 11, and 12 (and elsewhere in the manuscript), please consider including different plots, which would help elucidate the claims made

in the text more clearly.

We do not believe that difference plots would help the reader much. As the reviewer already mentioned, precipitation differences are fairly obvious in Fig. 10. At that time of convective stage, there are no differences in CAPE between individual model runs (only after 1200 UTC differences can be seen as a cause of convection/no convection). The important point is the location of the cell which is more north in areas of lower CAPE for the unsuccessful runs. Figure 12 is a collection of plots from the same model run for different times, so difference plots cannot be applied here.

**Changes to paper:**
none

**Minor/Specific Comments**

- Line 33, Line 36: What is the convective event on 8 June? The only mention of deep convection is the MCS on 9 June (Line 27). Please consider highlighting the preceding day or remove mention of an event on 8 June.

  We agree with the reviewer that the events on June 8 are not essential for the reader and removed the results but kept the mentioning of the day.

  **Changes to paper:**
  A series of different numerical simulations for the convective events of  9 June 2014 *and the previous day* were performed with the COSMO model, the main findings were

  – The COSMO model (in quasi operational set-up, without data assimilation) initialized at 00:00 UTC did not reproduc the mesoscale convective system (MCS) on 9 June.

  – The enlargement of the model domain towards the West  *improved the precipitation forecast over France only* due to better resolving the initiation and development of deep convection over western France and, later, secondary initiation over northern France. *The MCS over Germany, however, was not simulated even with this larger domain.*

  –

- Lines 49-51: But this increase in lead time was not observed in the simulations by Barthlott et al. (2017), correct? Lines 37-38 only mention that extending the model domain "had the largest effect". Please clarify what "largest effect" means in relation to the forecast skill and if this skill improvement corroborates the statement in Lines 49-51.

  The increase in lead time mentioned in lines 49-51 is just of theoretical nature if we assume that a convection-permitting model runs over a larger domain and successfully simulates the convection to come. In the study of Barthlott et al. (2017), the enlargement of the domain had a positive impact only for the preceding day and not for the day with the MCS discussed here. We modified the text to make that clearer.

  **Changes to paper:**
  The enlargement of the model domain towards the West  *improved the precipitation forecast over France only* due to better resolving the initiation and development of deep convection over western France and, later, secondary initiation over northern France. *The MCS over Germany on 9 June 2014, however, was not simulated even with this larger domain.*

- Line 55: "there are various to generate"

We believe that our phrasing is correct here: *"There a various ways of generating an ensemble..."*

**Changes to paper:**
none

- Line 59: "Recent studies by"

  done

- Lines 59-61: How do these studies contribute to ensemble generation (Line 56)? These references seem out of place for what this section is discussing.

  There are various ways of generating an ensemble and some of them are cited in that parapgraph. The studies on lines 59-61 address other and new approaches in ensemble modelling and are therefore mentioned here. Please note that these lines are now in section 5.

  **Changes to paper:**
  none

- Lines 7,94,182: I urge the authors to consider removing "surprising" and "surprisingly" from the manuscript. The results are NOT surprising when you consider the domain shifting technique as a simple stochastic perturbation, which has been shown to have large impacts on convection by others when considering perturbed lateral boundary conditions (e.g., Clark et al. 2010, Romine et al. 2014). This wording is much too subjective and not scientific. Moreover, the general variability of convective systems can be dramatic within convection-allowing models, as demonstrated in other studies (e.g., Melhauser and Zhang 2012), so I would not consider the current results "surprising".

  We do believe that our technique of domain shifting of just 1 grid point provides surprising or at least unexpected results for this particular case. Given the large model domain and the minor changes at the boundaries, we would not have anticipated such a large dependency. However, we follow the reviewer's suggestion and eliminate that wording.

  Nevertheless, we do believe that scientists can be surprised. A keyword search for "surprising" in the AMS journals of the last 10 years leads to more than 7000 hits.

  **Changes to paper:**
  Abstract and Introduction:
  Section 4.2 and Summary:  $\rightarrow$ *unexpected*

- Lines 109-110: What initiated the storms during the night and morning hours? Shortwave trough?

  Please see our reply to the fourth subsequent remark. "The showers near Nantes..."

- Line 129: Please specify the date with the initialization time

  done

- Line 148: I think this is the first time the authors have mentioned "reference run". I would suggest highlighting the reference/control simulation in the previous section, maybe stating that before any domain shifting occurs, you run a reference/control simulation of the event to compare shifted experiments against. Section 4 then naturally transitions to the discussion of reference forecast skill.

  This is a good suggestion. We now introduce the reference run already in section 3.2.

**Changes to paper:**
Section 3.2:
*At first, we conducted a reference run (REF) with a model domain containing*  600×650 grid points, which corresponds to an area of about 1668 km×1807 km.

We also included the reference run in Table 1.
Table 1. *Domain shifting*

Section 4.1:

Here we compare our *reference* simulation...

- Line 159: "of of" to "of"

  done

- Lines 169-170: What initiated the showers near Nantes? What was the initiating mechanism that ultimately led to the extreme event in Germany later in the forecast period?

  The showers near Nantes during the night were initiated by a combination of large-scale forcing and low-level wind convergence. Fig. R.1 shows low-level winds and Q-vector divergence between 0100 UTC and 0230 UTC. Although CAPE is absent, there is a region with Q-vector divergence (negative values) indicating forcing for upward vertical motion which coincides with low-level wind convergence. We added additional information in the text.

  Later on, more weak showers formed near the coast which traveled towards northern France. One of these showers then developed into the MCS as sufficient CAPE was present in the afternoon and the wind shear favoured organized long-lived convection by outflow triggering. The proximity of the track of the convective system to the north coast of France and the colder temperatures over the sea than the land then determine the large differences in the model results. These processes are analyzed in detail in section 4.6.

  **Changes to paper:**
  *"The precursors of these showers were initiated over the sea by a combination of large-scale forcing (determined by Q-vector divergence) and low-level wind convergence (not shown)."*

- Line 170: How was this track determined? Subjectively? Please specify

  Yes, the track was subjectively determined. We included that information in the text.

  **Changes to paper:**
  The *subjectively determined* track of the system...

- Line 174: "too far north"

  done

  **Changes to paper:**
   *too far north*

- Lines 184-187: These two sentences are at odds with one another. The first sentence states that there are no systematic responses to the forecasts when shifting the domain in a specific direction. The second sentence actually describes a systematic response resulting in poor forecasts when shifting to the east. Please rectify this discrepancy. Please switch sections 4.2 and 4.3. It makes more sense to discuss the reference forecast, the sensitivity to domain choice, and then the results of the experiment simulations. It reads disjointed as is.

[Figure]

Figure R.1: Left: CAPE (colour shading), precipitation (blue colour shading), and 10-m wind field (arrows); Right: Q-vector divergence (colours), 500 hPa geopotential height (contour lines), and precipitation rate (hatched). From top to bottom: 0100, 0130, 0200, 0230 UTC.

We deleted the second sentence as the E run did produce precipitation over Germany, but somewhat shifted northwards. The statement from the first sentence is valid, i.e. that there is no systematic response to the forecasts when shifting the domain in a specific direction.

We believe that it is not necessary to switch sections 4.2 and 4.3. We first want to show the sensitivity of convective precipitation to domain choice (4.2) and then find possible reasons for this behavior (4.3 Differences in IC, 4.4 Convection-related parameters,...). We therefore decided to keep the sections in this order.

**Changes to paper:**

- Lines 210-211: Please reference relevant literature on this topic (e.g., Ancell et al. 2018 and others) to support your claim.

We decided to remove that sentence, as it is rather speculative and we show later that small

displacements of the precursor cell determines whether a large MCS develops or not.

**Changes to paper:**

- Line 219: Please illustrate this box in one of your figures. A reader should be able to see/visualize where this box is placed at a particular time for the analysis presented in this section.

As we already depicted in the text, the box follows the storm along the path depicted in Fig. 6. So for different times, the box has different locations. Moreover, the location of the box would only be meaningful for 30-min rain rates and not for accumulated precipitation. It is not feasible to display all boxes for all times in the same figure (Fig. R.2):

[Figure]

Figure R.2: 24-h accumulated precipitation with boxes for all times (left) and every 6 hours (right).

**Changes to paper:**
We included the location of 4 boxes corresponding to four times (06:00, 12:00, 18:00, 24:00 UTC) in Fig. 6e.

- Line 225: I do not see the "slight reduction in intensity" within the Fig. 8a black line at 1700 UTC.

You are right, we re-phrased that sentence.

**Changes to paper:**
*"The precipitation rate of the REF run is gradually increasing until 14:30 UTC (Fig. 8a) and remains more or less constant until 17:00 UTC. Then precipitation rate is strongly increasing until 18:30 UTC followed by a slight reduction in intensity before the maximum is reached between 21:00 UTC and 22:00 UTC."*

- Line 225: Between 2100 and 2200 UTC

done

- Lines 231-234: There is a clear bifurcation in directional shear between good and bad experiment forecasts by 1300 UTC. Can the authors please discuss the possible implications of these directional shear differences as they relate to MCS development and propagation?

Both deep layer shear and directional shear are similar in all model runs until 12:00 UTC and show suitable conditions for highly-organized convection. It is only after convection has intensified in the successful runs (12:00–13:00 UTC), that there is a bifurcation of directional shear and speed shear between the good and bad experiment forecasts. Therefore we believe that these differences are a result of the (non)-existent MCS itself and not the driver for it. In all model runs, DLS is higher than 20 m/s until 21:00 UTC which shows the potential for organized convection for all model runs. But because of the processes discussed later in the paper (initial convection too close to colder sea), not all runs simulate the MCS. We further comment on that later in a similar question.

**Changes to paper:**
none

- Lines 245-246: This is a reasonable assumption...but can the authors actually prove this is what is occurring and not some other mesoscale process?

Our assumption that precipitation is consuming CAPE is further outlined below in similar questions (Lines 265-266/Line 269) to that topic.

**Changes to paper:**
none

- Line 250: What is the implication of reference simulation and "unsuccessful" runs having similar proportions of CAPE-to-CIN? It appears to me that the absence of CIN is the major contributor to initiation for the "successful" (warmer colors) runs.

[Figure]

Figure R.3: Time series of domain-averaged CIN.

CIN or the absence of CIN is not the major contributor for dividing into successful and not successful runs. We computed domain-averaged CIN (Fig. R.3). It can be seen that until 1100–1200 UTC, CIN is on average similar in all model runs. Differences in CIN emerge between 1300–1600 UTC when convective precipitation is intensifying. We added a remark in the paper.

**Changes to paper:**
*"The fraction of grid points fulfilling that criterion is primarily dominated by the existence of CAPE as domain-averaged CIN is very similar in all model runs until 12:00 UTC (not shown)."*

- Lines 265-266: Can the authors show this is indeed the cause of lower CAPE and shear in the successful runs? It reads as speculative and should be backed up with some analysis, at the very least shown to the reviewers

Please see our response to the next question, which covers the same topic.

- Line 269: Can the authors please provide supporting evidence that indeed the lower CAPE and shear is a result of storm modification? This appears to be speculation, and should be corroborated with evidence.

  As can be seen in the time series of Fig. 8, CAPE is building up from the early morning hours in all model runs. The runs without the MCS simulate an increase in CAPE until 1700–1800 UTC, after that CAPE is reduced due to a reduction of near-surface temperatures. The precipitation in the successful runs, however, consume that CAPE. The model also simulates a time lag of 2-3 h between the CAPE and precipitation with former leading the latter, which is typical for convection over land. It is also evident from Fig. 10 that convective precipitation consumes CAPE, as in the wake of the MCS, CAPE is significantly reduced. This is also evident from Fig. R.4 in which precipitation is displayed as a function of CAPE. The not successful W run simulates an increase in CAPE until 1930 UTC, then CAPE is reduced due to the reduction of near-surface temperatures. The reference run, on the other hand, simulates an increase in CAPE until 1730 UTC only. Then, CAPE is consumed by the precipitation of the convective system. We believe that this is enough evidence for the cause of lower CAPE in the successful runs.

[Figure]

Figure R.4: Precipitation vs. CAPE of the W run and the reference run. Simulation time is indicated by the numbers inside the dots.

  The reduced deep layer shear is similar in all runs until 1300 UTC. After that time, the time series of precipitation diverge into the successful runs (increasing precipitation) and the not successful runs (decreasing precipitation). Such a separation is also visible in the time series of deep layer shear with the successful runs having lower values. We believe that the temporal evolution is evidence enough for the fact that the MCS convection modifies its environment and not vice versa.

  **Changes to paper:**
  none

- Lines 288-290: Why not compute a sensitivity analysis to the fields listed in Figure 8?

  Figure 9 now does show the sensitivity to some of the fields shown in Figure 8. Showing all fields would be repetitive - see the longer reply above.

  **Changes to paper:**

Figure 9 now does show the sensitivity to some of the fields shown in Figure 8. The text in section 4.4 is changed accordingly.

- Line 290: Please show this box in a figure

  **Changes to paper:**
  Box added to Figure 6e.

[revised manuscript text omitted]